# Regulation of hippocampal mossy fiber-CA3 synapse function by a Bcl11b/C1ql2/Nrxn3(25b+) pathway

**Artemis Koumoundourou[1], Märt Rannap[2], Elodie De Bruyckere[1], Sigrun Nestel[3], Carsten Reissner[4], Alexei V Egorov[2], Pengtao Liu[5,6], Markus Missler[4], Bernd Heimrich[3], Andreas Draguhn[2], Stefan Britsch[1]\***

[1]Institute of Molecular and Cellular Anatomy, Ulm University, Ulm, Germany; [2]Institute of Physiology and Pathophysiology, Faculty of Medicine, Heidelberg University, Heidelberg, Germany; [3]Department of Neuroanatomy, Institute of Anatomy and Cell Biology, Faculty of Medicine, University of Freiburg, Freiburg, Germany; [4]Institute of Anatomy and Molecular Neurobiology, University of Münster, Münster, Germany; [5]School of Biomedical Sciences, Li Ka Shing Faculty of Medicine, The University of Hong Kong, Hong Kong, China; [6]Centre for Translational Stem Cell Biology, Hong Kong, China

**\*For correspondence:**
stefan.britsch@uni-ulm.de

**Competing interest:** The authors declare that no competing interests exist.

**Abstract** The transcription factor Bcl11b has been linked to neurodevelopmental and neuropsychiatric disorders associated with synaptic dysfunction. Bcl11b is highly expressed in dentate gyrus granule neurons and is required for the structural and functional integrity of mossy fiber-CA3 synapses. The underlying molecular mechanisms, however, remained unclear. We show in mice that the synaptic organizer molecule C1ql2 is a direct functional target of Bcl11b that regulates synaptic vesicle recruitment and long-term potentiation at mossy fiber-CA3 synapses in vivo and in vitro. Furthermore, we demonstrate C1ql2 to exert its functions through direct interaction with a specific splice variant of neurexin-3, Nrxn3(25b+). Interruption of C1ql2-Nrxn3(25b+) interaction by expression of a non-binding C1ql2 mutant or by deletion of Nrxn3 in the dentate gyrus granule neurons recapitulates major parts of the Bcl11b as well as C1ql2 mutant phenotype. Together, this study identifies a novel C1ql2-Nrxn3(25b+)-dependent signaling pathway through which Bcl11b controls mossy fiber-CA3 synapse function. Thus, our findings contribute to the mechanistic understanding of neurodevelopmental disorders accompanied by synaptic dysfunction.

## eLife assessment

The authors identify a new role for C1ql2 at mossy fiber synapses in the hippocampus and **convincingly** find that C1ql2, whose expression is controlled by Bcl11b, controls the recruitment of synaptic vesicles to active zones and is necessary for synaptic plasticity. These **important** results build upon prior discoveries of how Bcl11b, a disease-relevant molecule, contributes to our understanding of mossy-fiber synaptic development.

## Introduction

Disruptions in synaptic structure and function have been identified as a major determinant in the manifestation of various neurodevelopmental and neuropsychiatric disorders, such as autism spectrum disorder, schizophrenia, and intellectual disability (*Hayashi-Takagi, 2017*; *Lepeta et al., 2016*; *Zoghbi and Bear, 2012*). The need to understand how synaptic function is compromised in these disorders

**eLife digest** The human brain contains billions of neurons working together to process the vast array of information we receive from our environment. These neurons communicate at junctions known as synapses, where chemical packages called vesicles released from one neuron stimulate a response in another. This synaptic communication is crucial for our ability to think, learn and remember.

However, this activity depends on a complex interplay of proteins, whose balance and location within the neuron are tightly controlled. Any disruption to this delicate equilibrium can cause significant problems, including neurodevelopmental and neuropsychiatric disorders, such as schizophrenia and intellectual disability.

One key regulator of activity at the synapse is a protein called Bcl11b, which has been linked to conditions affected by synaptic dysfunction. It plays a critical role in maintaining specific junctions known as mossy fibre synapses, which are important for learning and memory. One of the genes regulated by Bcl11b is C1ql2, which encodes for a synaptic protein. However, it is unclear what molecular mechanisms Bcl11b uses to carry out this role.

To address this, Koumoundourou et al. explored the role of C1ql2 in mossy fibre synapses of adult mice. Experiments to manipulate the production of C1ql2 independently of Bcl11b revealed that C1ql2 is vital for recruiting vesicles to the synapse and strengthening synaptic connections between neurons. Further investigation showed that C1ql2's role in this process relies on interacting with another synaptic protein called neurexin-3. Disrupting this interaction reduced the amount of C1ql2 at the synapse and, consequently, impaired vesicle recruitment.

These findings will help our understanding of how neurodevelopmental and neuropsychiatric disorders develop. Bcl11b, C1ql2 and neurexin-3 have been independently associated with these conditions, and the now-revealed interactions between these proteins offer new insights into the molecular basis of synaptic faults. This research opens the door to further study of how these proteins interact and their roles in brain health and disease.

has accentuated the importance of studying the regulatory mechanisms of physiological synaptic function. These mechanisms involve cell adhesion molecules at both the pre- and post-synaptic side that act as synaptic organizers, whose unique combination determines the structural and functional properties of the synapse. Many such proteins have already been identified and our understanding of their complex role in synapse assembly and function has significantly increased over the last years (*de Wit and Ghosh, 2016*; *O'Rourke et al., 2012*; *Südhof, 2017*). Furthermore, recent advances in the genetics of neurodevelopmental and neuropsychiatric disorders have implicated genes encoding for several of the known synaptic proteins, supporting a role for these molecules in the pathogenesis of corresponding disorders (*Südhof, 2021*; *Torres et al., 2017*; *Wang et al., 2018*).

The sensitivity of the functional specification of the synapse to the combination of distinct synaptic proteins and their relative expression levels suggests that genetically encoded programs define at least facets of the synaptic properties in a cell-type-specific manner (*Südhof, 2017*). Several of the synaptic proteins have been shown to promote formation of functional pre- and postsynaptic assemblies when presented in non-neuronal cells (*Dalva et al., 2000*; *Dean et al., 2003*; *Scheiffele et al., 2000*), showing that their ability to specify synapses is in part independent of signaling processes and neuronal activity and supporting the idea that synaptic function is governed by cues linked to cellular origin (*Gomez et al., 2021*). Thus, the investigation of synaptic organizers and their function in health and disease should be expanded to the transcriptional programs that regulate their expression.

Bcl11b (also known as Ctip2) is a zinc finger transcription factor that has been implicated in various disorders of the nervous system including Alzheimer's and Huntington's disease, and schizophrenia (*Kunkle et al., 2016*; *Song et al., 2022*; *Whitton et al., 2018*; *Whitton et al., 2016*). Patients with *BCL11B* mutations present with neurodevelopmental delay, overall learning deficits as well as impaired speech acquisition and autistic features (*Eto et al., 2022*; *Lessel et al., 2018*; *Punwani et al., 2016*; *Yang et al., 2020*). Bcl11b is expressed in several neuron types, including the dentate gyrus granule neurons (DGN) of the hippocampus. Expression of Bcl11b in the DGN starts during embryonic development and persists into adulthood (*Simon et al., 2020*). We have previously demonstrated that Bcl11b plays a crucial role in the development of the hippocampal mossy fiber system,

adult hippocampal neurogenesis as well as hippocampal learning and memory (*Simon et al., 2016*; *Simon et al., 2012*). In the mature hippocampus, Bcl11b is critical for the structural and functional integrity of mossy fiber synapses (MFS), the connections between DGN and CA3 pyramidal neurons (*De Bruyckere et al., 2018*). MFS have a critical role in learning and memory stemming from their unique structural and functional properties, such as an enormous pool of releasable synaptic vesicles (SV), and reliable presynaptic short- and long-term plasticity (*Nicoll and Schmitz, 2005*; *Rollenhagen and Lübke, 2010*). Conditional ablation of *Bcl11b* in murine DGN impairs presynaptic recruitment of SV and abolishes mossy fiber long-term potentiation (MF-LTP; *De Bruyckere et al., 2018*). The molecular mechanisms, however, through which the transcriptional regulator Bcl11b controls highly dynamic properties of the MFS remained elusive.

In the present study, we show that the secreted synaptic organizer molecule C1ql2, a member of the C1q-like protein family (*Yuzaki, 2017*), is a functional target of Bcl11b in murine DGN. Reintroduction of C1ql2 in *Bcl11b* mutant DGN rescued the localization and docking of SV to the active zone (AZ), as well as MF-LTP that was abolished upon *Bcl11b* ablation. Knock-down (KD) of *C1ql2* in wild-type animals recapitulated a major part of the MFS phenotype observed in *Bcl11b* mutants. Furthermore, we show that C1ql2 requires direct interaction with a specific neurexin-3 isoform, Nrxn3(25b+), a member of a polymorphic family of presynaptic cell adhesion molecules (*Reissner et al., 2013*; *Südhof, 2017*), to recruit SV in vitro and in vivo. Finally, we observe that localization of C1ql2 along the mossy fiber tract depends on C1ql2-Nrxn3(25b+) interaction. Taken together, this study identifies a novel Bcl11b/C1ql2/Nrxn3(25b+)-dependent regulatory mechanism that is essential for the control of MFS function. Recent genetic studies suggested its single components to be associated with neurodevelopmental and neuropsychiatric disorders characterized by synaptic dysfunction. Our data, for the first time, demonstrate these molecules to be interconnected in one regulatory pathway. Thus, our findings provide new mechanistic insight into the pathogenesis of corresponding human disorders.

## Results

### Reintroduction of C1ql2 into Bcl11b mutant dentate granule neurons restores synaptic vesicle recruitment at the mossy fiber-CA3 synapse

We demonstrated before that Bcl11b is critical for the structural and functional integrity of adult excitatory hippocampal MFS (*De Bruyckere et al., 2018*; *Simon et al., 2016*). The downstream regulatory mechanisms, however, through which Bcl11b exerts its complex functions at the MFS remained unclear. In a previous study, we carried out differential transcriptomic analyses on *Bcl11b* conditional knock-out (cKO) and wildtype (WT) DGN, to systematically screen for candidate transcriptional targets of Bcl11b (*De Bruyckere et al., 2018*). Among the differentially expressed candidate genes, we identified the synaptic organizer molecule C1ql2 (*De Bruyckere et al., 2018*), previously implicated in modulating MFS functions (*Matsuda et al., 2016*). *C1ql2* transcript and protein levels are massively downregulated in *Bcl11b* mutant DGN (*Figure 1—figure supplement 1*; *De Bruyckere et al., 2018*), and the Bcl11b protein directly binds to consensus sequences within the *C1ql2* promotor (*De Bruyckere et al., 2018*), suggesting Bcl11b to act on MFS through C1ql. To directly test this, we stereotaxically injected a C1ql2-expressing AAV (*Figure 1a–b*) into the dentate gyrus (DG) of *Bcl11b* cKO mice 2 weeks after induction of the mutation and compared them to control animals. To avoid potential interference of the AAV-mediated gene expression with the interpretability of observed phenotypes, we stereotaxically injected the DG of control animals as well, with AAV expressing EGFP only. AAV-mediated re-expression of *C1ql2* in the *Bcl11b* mutant DGN completely restored C1ql2 protein expression (*Figure 1c–d*; Control +EGFP: 1±0.216, Bcl11b cKO +EGFP: 0.2±0.023, Bcl11b cKO +EGFP-2A-C1ql2: 2.44±0.745, mean ± SEM). Furthermore, the spatial distribution of the exogenous C1ql2 protein in mutants was indistinguishable from controls (*Figure 1e*). Using vGlut1 and Homer1 as pre- and postsynaptic markers, respectively, we observed exogenous C1ql2 protein to precisely localize at glutamatergic synapses within the stratum lucidum (SL) of CA3, confirming that reintroduced C1ql2 is correctly targeted to the MFS (*Figure 1f*).

MFS of *Bcl11b* cKO animals were characterized by a misdistribution of SV in relation to the AZ, with fewer SV being present in the vicinity of AZ, as reflected by a lower average synapse score (*Figure 2a–b*). The scoring system used in this study rates MFS based on the number of SV and their

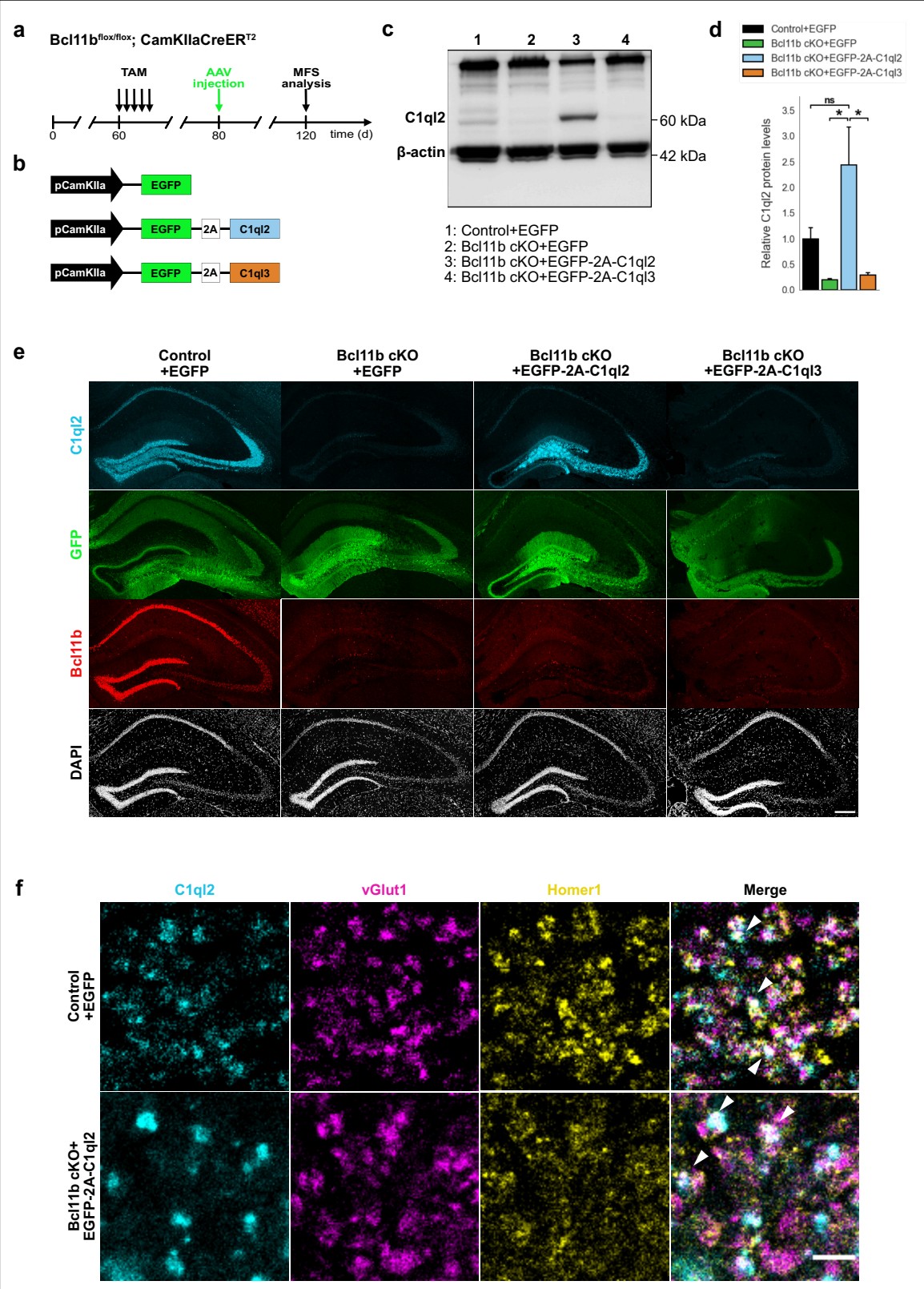

**Figure 1.** Stereotaxic injection of C1ql2-expressing AAV into *Bcl11b* cKO DGN restores C1ql2 levels. (**a**) Experimental design to analyze the functions of C1ql2 in the MFS as a downstream target of Bcl11b. (**b**) AAV constructs injected in the DG of *Bcl11b* cKO and control littermates. (**c**) Western blot and (**d**) relative C1ql2 protein levels in mouse hippocampal homogenates. N=3. All data are presented as means; error bars indicate SEM. Two-way ANOVA and Tuckey's PHC. Control +EGFP vs. Bcl11b cKO +EGFP-2A-C1ql2: ns, p=0.11; Bcl11b cKO +EGFP-2A-C1ql2 vs. Bcl11b cKO +EGFP: *p=0.015; Bcl11b

*Figure 1 continued on next page*

*Figure 1 continued*

cKO +EGFP-2A-C1ql2 vs. Bcl11b cKO +EGFP-2A-C1ql3: *p=0.019; ns, not significant. (**e**) Immunohistochemistry of C1ql2 (cyan), GFP (green), and Bcl11b (red) on hippocampal sections. Scale bar: 200 µm. (**f**) Immunohistochemistry of C1ql2 (cyan), vGlut1 (magenta), and Homer1 (yellow) in the SL of CA3. White arrowheads indicate co-localizing puncta of all three proteins. Scale bar: 15 µm.

The online version of this article includes the following source data and figure supplement(s) for figure 1:

**Source data 1.** File containing the raw data for *Figure 1*, panel d.

**Source data 2.** Original file for the western blot analysis in *Figure 1c*.

**Source data 3.** PDF containing *Figure 1c* and original scans of the relevant western blot analysis with highlighted bands and sample labels.

**Figure supplement 1.** C1ql2 mRNA and protein are lost upon *Bcl11b* cKO in DGN.

distance from the AZ, is described in detail in the Materials and methods part, and has been published previously (*De Bruyckere et al., 2018*). Reintroduction of C1ql2 fully recovered the synapse score to control values (*Figure 2a–b*; Control +EGFP: 3.4±0.012, Bcl11b cKO +EGFP: 2.96±0.037, Bcl11b cKO +EGFP-2A-C1ql2: 3.47±0.043, mean ± SEM). As revealed by the relative frequency of the individual synapse scores, C1ql2 not only reduced the number of inactive synapses, characterized by a synapse score of 0, but also improved the synapse score of active synapses (*Figure 2c*). To test for the specificity of the C1ql2 effects, we overexpressed C1ql3 in the DGN of *Bcl11b* cKO (*Figure 1a–b*). C1ql3, a different member of the C1ql subfamily, is co-expressed with C1ql2 in DGN and the two proteins have been shown to form functional heteromers (*Matsuda et al., 2016*). C1ql3 expression is unchanged in *Bcl11b* cKO (*Figure 2—figure supplement 1a-b*; Control +EGFP: 1±0.022, Bcl11b cKO +EGFP: 1.09±0.126, Bcl11b cKO +EGFP-2A-C1ql2: 0.87±0.146, mean ± SEM). Overexpression of C1ql3 in the DGN of *Bcl11b* cKO neither interfered with C1ql2 expression levels (*Figure 1c–e*; Bcl11b cKO +EGFP-2A-C1ql3: 0.29±0.042, mean ± SEM) nor was it able to rescue the synapse score of *Bcl11b* mutants (*Figure 2a–c*; Bcl11b cKO +EGFP-2A-C1ql3: 2.97±0.062, mean ± SEM). While not significant, AAV-mediated re-expression of C1ql2 in *Bcl11b* cKO led to artificially elevated C1ql2 protein levels compared to controls (*Figure 1c–d*). To exclude that the observed effects were influenced by the elevated C1ql2 expression in the *Bcl11b* cKO background above physiological levels, we over-expressed C1ql2 as well in control animals, which resulted in a strong increase of C1ql2 (*Figure 2h*). However, this did not affect the average synapse score (*Figure 2i–j*; Control +EGFP: 3.4±0.012; Control +EGFP-2A-C1ql2: 3.41±0.031; Bcl11b cKO +EGFP-2A-C1ql2: 3.47±0.043, mean ± SEM).

To analyze the C1ql2-dependent functions of Bcl11b on SV distribution in more detail, we quantified the number of SV docked on AZ in control animals, *Bcl11b* mutants, and upon the reintroduction of C1ql2. SV with a≤5 nm distance from the plasma membrane were considered docked (*Kusick et al., 2022*; *Vandael et al., 2020*). *Bcl11b* mutant animals had significantly fewer docked vesicles per 100 nm of AZ profile length compared to control animals and more AZ with no docked vesicles at all. Rescue of C1ql2 expression restored the number of docked SV to control levels, while the overexpression of C1ql3 did not affect the number of docked vesicles (*Figure 2d and f–g*; Control +EGFP: 0.53±0.098, Bcl11b cKO +EGFP: 0.24±0.038, Bcl11b cKO +EGFP-2A-C1ql2: 0.51±0.049, Bcl11b cKO +EGFP-2A-C1ql3: 0.26±0.041, mean ± SEM). The AZ length remained unchanged in all conditions (*Figure 2—figure supplement 1c*; Control +EGFP: 168.3±4,94, Bcl11b cKO +EGFP: 161.9±5.56, Bcl11b cKO +EGFP-2A-C1ql2: 161.91±7.14, Bcl11b cKO +EGFP-2A-C1ql3: 171.87±6.74, mean ± SEM). In contrast to vesicle docking, Bcl11b and C1ql2 did not affect the size of the docked SV (*Figure 2e*; Control +EGFP: 36.30±1.67, Bcl11b cKO +EGFP: 35.18±1.13, Bcl11b cKO +EGFP-2A-C1ql2: 36.35±1.01, Bcl11b cKO +EGFP-2A-C1ql3: 36.65±0.1, mean ± SEM). Thus, our data suggest that C1ql2 specifically controls SV recruitment downstream of Bcl11b.

Conditional deletion of *Bcl11b* in the adult hippocampus also leads to a loss of MFS, as well as reduced ultrastructural complexity of the remaining mossy fiber boutons (MFB; *De Bruyckere et al., 2018*), posing the question of whether these phenotypic features also depend on C1ql2. Interestingly, C1ql2 reintroduction in the DGN of *Bcl11b* cKO neither restored the loss of glutamatergic synapses, as quantified by the colocalization of pre- and postsynaptic markers, vGlut1 and Homer1 (*Figure 2—figure supplement 1d*; Control +EGFP: 90.65±8.25, Bcl11b cKO +EGFP: 60.68±4.62, Bcl11b cKO +EGFP-2A-C1ql2: 56.84±6.99, mean ± SEM), nor the reduced ultrastructural complexity of MFB, as quantified by the MFB perimeter/area ratio (*Figure 2—figure supplement*

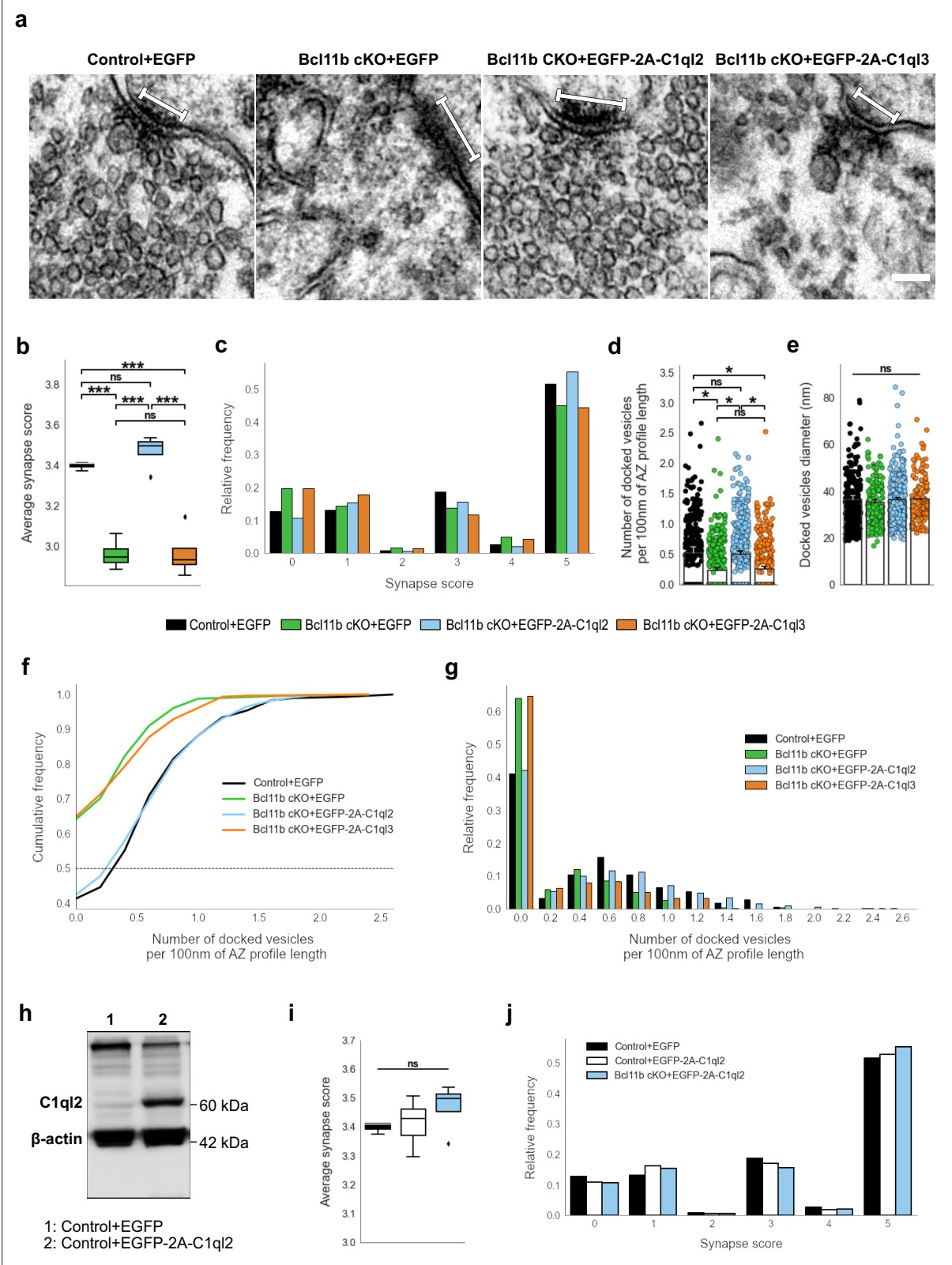

**Figure 2.** C1ql2 reintroduction in *Bcl11b* cKO DGN rescues SV recruitment in MFS. (**a**) Electron microscope images of MFS and proximal SV. White bars mark synapse length from the postsynaptic side. Scale bar: 100 nm. (**b**) Average synapse score. Control +EGFP, N=3; Bcl11b cKO +EGFP, Bcl11b cKO +EGFP-2A-C1ql2, Bcl11b cKO +EGFP-2A-C1ql3, N=4. Two-way ANOVA and Tuckey's PHC. Control +EGFP vs. Bcl11b cKO +EGFP: ***p=0.0002, and vs. Bcl11b cKO +EGFP-2A-C1ql3: ***p=0.0003; Bcl11b cKO +EGFP-2A-C1ql2 vs. Bcl11b cKO +EGFP and vs. Bcl11b cKO +EGFP-2A-C1ql3:

*Figure 2 continued on next page*

Figure 2 continued

***p<0.0001; ns, not significant. (**c**) Relative frequency of synapse scores. (**d**) Number of docked vesicles per 100 nm AZ profile length. Control +EGFP, Bcl11b cKO +EGFP-2A-C1ql3, N=3; Bcl11b cKO +EGFP, Bcl11b cKO +EGFP-2A-C1ql2, N=4. All data are presented as means; error bars indicate SEM. Points represent the individual examined AZ and SV, respectively. Two-way ANOVA and Tuckey's PHC. Control +EGFP vs. Bcl11b cKO +EGFP: *p=0.024, and vs. Bcl11b cKO +EGFP-2A-C1ql3: *p=0.045; Bcl11b cKO +EGFP-2A-C1ql2 vs. Bcl11b cKO +EGFP: *p=0.026, and vs. Bcl11b cKO +EGFP-2A-C1ql3: *p=0.049; ns, not significant. (**e**) Diameter of docked vesicles. Control +EGFP, Bcl11b cKO +EGFP-2A-C1ql3, N=3; Bcl11b cKO +EGFP, Bcl11b cKO +EGFP-2A-C1ql2, N=4; Two-way ANOVA. ns, not significant. (**f**) Cumulative and (**g**) relative frequency of the number of docked vesicles per 100 nm AZ profile length. (**h**) Western blot of mouse hippocampal homogenates. (**i**) Average synapse score. Control +EGFP, N=3; Control +EGFP-2A-C1ql2, N=6; Bcl11b cKO +EGFP-2A-C1ql2, N=4. Two-way ANOVA. ns, not significant. (**j**) Relative frequency of synapse scores. Data for Control +EGFP-2A-C1ql2 from i-j in this figure are compared with Control +EGFP and Bcl11b cKO +EGFP-2A-C1ql2 data from (**b-c**).

The online version of this article includes the following source data and figure supplement(s) for figure 2:

Source data 1. File containing the raw data for *Figure 2 b-g and i-j* and for *Figure 2—figure supplement 1b-d and f*.

Source data 2. Original file for the western blot analysis in *Figure 2h*.

Source data 3. PDF containing *Figure 2h* and original scans of the relevant western blot analysis with highlighted bands and sample labels.

Figure supplement 1. C1ql2 reintroduction in *Bcl11b* cKO DGN does not rescue MFS number and MFB complexity.

*1e-f*; Control +EGFP: 0.0051±0.00031, Bcl11b cKO +EGFP: 0.0042±0.00014, Bcl11b cKO +EGFP-2A-C1ql2: 0.0037±0.00021, mean ± SEM). This suggests that Bcl11b acts on MFS through C1ql2-dependent as well as -independent signaling pathways.

## Reintroduction of C1ql2 into Bcl11b mutant dentate granule neurons rescues mossy fiber synapse long-term potentiation

The ultrastructural changes at the MFS point towards potential alterations in synaptic function. Indeed, adult-induced *Bcl11b* cKO was previously found to result in a loss of MF-LTP (*De Bruyckere et al., 2018*). We therefore tested whether the reintroduction of C1ql2 in *Bcl11b* cKO DGN can rescue LTP at the mutant MFS, similarly to SV recruitment. We stimulated mossy fibers in acute slices and measured the resulting field potentials in the SL of CA3. Field responses were carefully validated for the specificity of mossy fiber signals by the presence of strong paired-pulse facilitation and block by the mGluR antagonist DCG-IV. Under these conditions, input-output curves of fEPSP slopes versus axonal fiber volleys revealed no significant differences between control and *Bcl11b* cKO mice (*Figure 3—figure supplement 1a-b*; Control +EGFP: 1.95±0.09, Bcl11b cKO +EGFP: 2.02±0.14, mean ± SEM), indicating that basal synaptic transmission was unaltered in the *Bcl11b* mutants. We then induced LTP by high-frequency stimulation (HFS) of mossy fibers in control and *Bcl11b* cKO animals with or without AAV-mediated expression of C1ql2. Compared to controls, *Bcl11b* mutants injected with the control AAV displayed a strong reduction of LTP at 20–30 and 30–40 min after induction (*Figure 3a–c*; 0–10 min: Control +EGFP: 90.4±7.2, Bcl11b cKO +EGFP: 106.1±10.8, 10–20 min: Control +EGFP: 42.7±3.6, Bcl11b cKO +EGFP: 39.5±4.6, 20–30 min: Control +EGFP: 52.5±7.6, Bcl11b cKO +EGFP: 24.8±3.2, 30–40 min: Control +EGFP: 50.1±7.3, Bcl11b cKO +EGFP: 20.3±3.7, mean ± SEM), consistent with our previous data (*De Bruyckere et al., 2018*). The loss of LTP was completely reversed upon the re-expression of C1ql2, with the rescue mice exhibiting comparable LTP to controls at all time intervals (*Figure 3a–c*; 0–10 min: Control +EGFP: 90.4±7.2, Bcl11b cKO +EGFP-2A-C1ql2: 86.5±7.4, 10–20 min: Control +EGFP: 42.7±3.6, Bcl11b cKO +EGFP-2A-C1ql2: 49.4±5.9, 20–30 min: Control +EGFP: 52.5±7.6, Bcl11b cKO +EGFP-2A-C1ql2: 47.2±5.7, 30–40 min: Control +EGFP: 50.1±7.3, Bcl11b cKO +EGFP-2A-C1ql2: 44.9±5.3, mean ± SEM). Importantly, this rescue effect was specific to C1ql2 as the overexpression of C1ql3 failed to reverse the *Bcl11b* cKO phenotype (*Figure 3a–c*; 0–10 min: Control +EGFP: 90.4±7.2, Bcl11b cKO +EGFP-2A-C1ql3: 104.2±9.9, 10–20 min: Control +EGFP: 42.7±3.6, Bcl11b cKO +EGFP-2A-C1ql3: 44.4±5.7, 20–30 min: Control +EGFP: 52.5±7.6, Bcl11b cKO +EGFP-2A-C1ql3: 29.0±2.3, 30–40 min: Control +EGFP: 50.1±7.3, Bcl11b cKO +EGFP-2A-C1ql3: 22.6±2.4, mean ± SEM).

MF-LTP is known to be mediated by the second messenger cAMP, which is produced by adenylyl cyclase (AC) in response to $Ca^{2+}$ influx through voltage-gated $Ca^{2+}$ channels (*Li et al., 2007*) and kainate receptors (KAR) (*Lauri et al., 2001*; *Schmitz et al., 2003*). To test whether Bcl11b acts on LTP by interfering with presynaptic $Ca^{2+}$ dynamics, we directly activated the cAMP pathway in slices from control and *Bcl11b* cKO mice by applying the AC activator forskolin (*Weisskopf et al., 1994*).

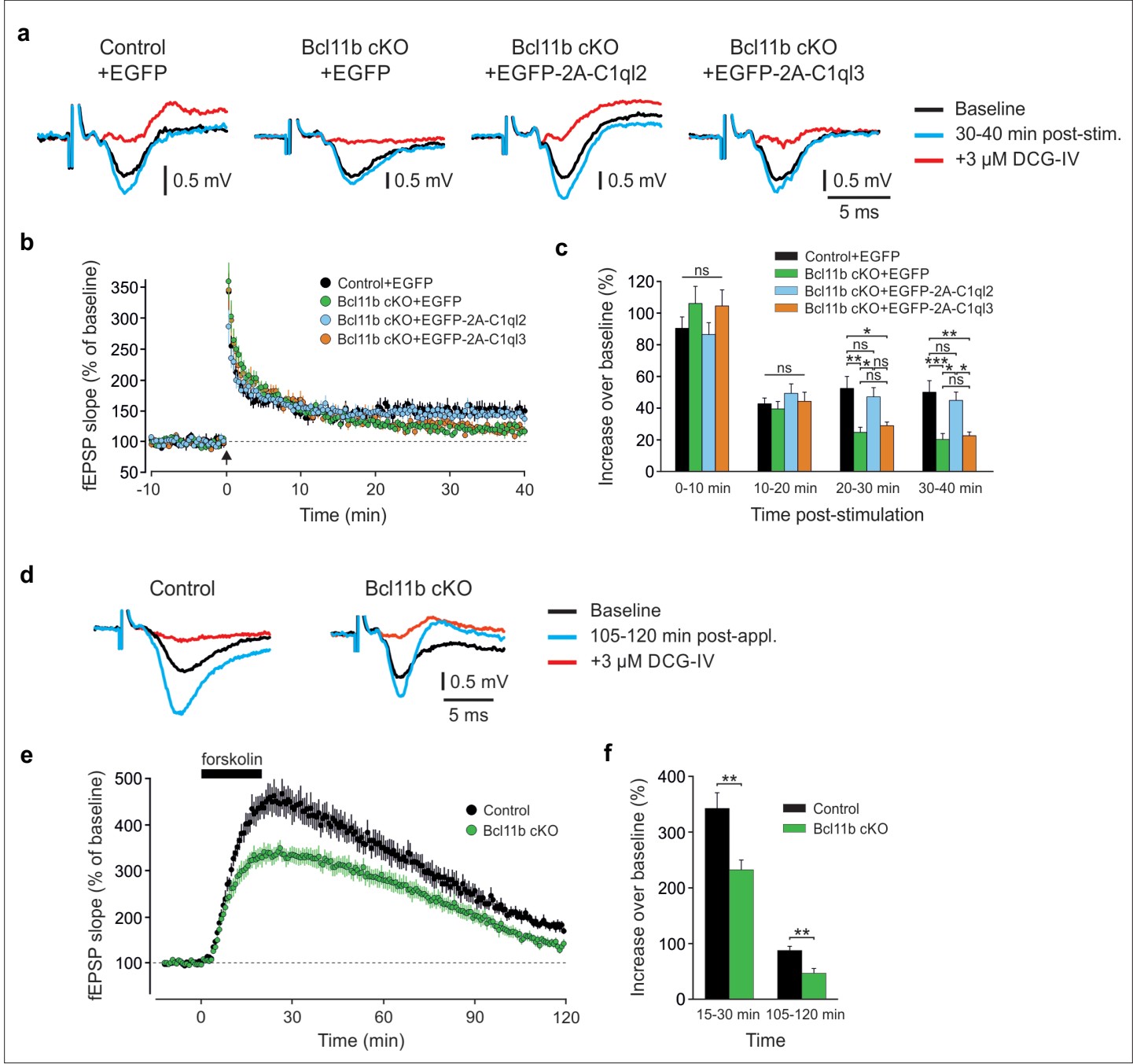

**Figure 3.** C1ql2 reintroduction in *Bcl11b* cKO DGN rescues mossy fiber LTP. (**a**) Representative fEPSP traces showing baselines before HFS (black), fEPSP changes 30–40 min after HFS (cyan) and following the application of 3 µM DCG-IV (red). (**b**) Time course of fEPSP slopes. The black arrow indicates HFS and the dashed line is the baseline level. (**c**) Quantification of fEPSP facilitation at four different time intervals after HFS. Changes in the fEPSP slope are shown as the percentage of the mean baseline fEPSP. Control +EGFP, 7 slices from 6 mice; Bcl11b cKO +EGFP, 8 slices from 5 mice, Bcl11b cKO +EGFP-2A-C1ql3, 8 slices from 6 mice; Bcl11b cKO +EGFP-2A-C1ql2, 6 slices from 4 mice; All data are presented as means; error bars indicate SEM. One-way ANOVA followed by Bonferroni's PHC for each time interval. 20–30 min: Control +EGFP vs. Bcl11b cKO +EGFP: **p=0.002, and vs. Bcl11b cKO +EGFP-2A-C1ql3: *p=0.011; Bcl11b cKO +EGFP-2A-C1ql2 vs. Bcl11b cKO +EGFP: *p=0.023; 30–40 min: Control +EGFP vs. Bcl11b cKO +EGFP: ***p<0.001, and vs. Bcl11b cKO +EGFP-2A-C1ql3: **p=0.002; Bcl11b cKO +EGFP-2A-C1ql2 vs. Bcl11b cKO +EGFP: *p=0.01 and vs. Bcl11b cKO +EGFP-2A-C1ql3: *p=0.023; ns, not significant. (**d**) Representative fEPSP traces showing baselines before forskolin application (black), fEPSP changes 105–120 min after the start of application (cyan) and following the addition of 3 µM DCG-IV (red). (**e**) Time course of fEPSP slopes. The black solid line indicates forskolin perfusion and the dashed line is the baseline level. (**f**) Quantification of fEPSP facilitation at two different time intervals after the start of the forskolin application. Changes in fEPSP slope are shown as percentage of the mean baseline fEPSP. 8 slices from 5 mice. All data are presented as means; error bars indicate SEM. Unpaired t-test for both time intervals. 15–30 min: **p=0.005; 105–120 min: **p=0.0025.

*Figure 3 continued on next page*

*Figure 3 continued*

The online version of this article includes the following source data and figure supplement(s) for figure 3:

**Source data 1.** File containing the raw data for *Figure 3*, panels b & e and for *Figure 3—figure supplement 1*, panels b and d.

**Figure supplement 1.** *Bcl11b* cKO and *C1ql2* KD in DGN do not affect basal synaptic transmission.

Compared to slices from control animals, forskolin-induced LTP in the mutants had a significantly lower peak and remained significantly weaker throughout the recording (*Figure 3d–f*; 15–30 min: Control: 342.4±28.0, Bcl11b cKO: 232.3±17.7, 105–120 min: Control: 88.0±7.4, Bcl11b cKO: 47.2±8.2, mean ± SEM). This suggests that the regulation of MF-LTP by Bcl11b involves the cAMP-dependent signaling pathway.

## Knock-down of C1ql2 in dentate granule neurons perturbs synaptic vesicle recruitment and long-term potentiation at the mossy fiber-CA3 synapse

To further corroborate the observation that Bcl11b acts on MFS specifically through C1ql2, we knocked down *C1ql2* expression in the DGN of adult WT mice by stereotaxically injecting an AAV carrying an shRNA cassette against *C1ql2* (*Figure 4a*). Quantitative PCR (*Figure 4b*), western blot analysis (*Figure 4c*), as well as immunohistochemistry using C1ql2 antibodies on hippocampal tissue (*Figure 4d*), revealed that the shRNA-mediated KD resulted in a strong reduction of *C1ql2* transcripts (*Figure 4b*;+shNS-EGFP: 1±0.07,+shC1ql2-EGFP: 0.23±0.059, mean ± SEM) as well as protein levels (*Figure 4c–d*), as compared to animals injected with the control AAV. The shRNA-mediated KD of *C1ql2* did not affect the expression of C1ql3, demonstrating the specificity of this approach (*Figure 4—figure supplement 1a-b*;+shNS-EGFP: 1±0.09,+shC1ql2-EGFP: 0.986±0.035, mean ± SEM). Compared to controls, *C1ql2* KD was sufficient to reduce the average synapse score to similar levels as observed in *Bcl11b* cKO (*Figure 4e*, *Figure 4—figure supplement 1c*;+shNS-EGFP: 3.38±0.069,+shC1ql2-EGFP: 3.15±0.031, mean ± SEM), as well as the number of docked vesicles per 100 nm of AZ profile length (*Figure 4f*, *Figure 4—figure supplement 1d-e*;+shNS-EGFP: 0.48±0.04,+shC1ql2-EGFP: 0.31±0.02, mean ± SEM). At the same time, the length of the AZ and the diameter of the docked vesicles remained unchanged (*Figure 4—figure supplement 1f-g*; AZ length:+shNS-EGFP: 172.96±8.24,+shC1ql2-EGFP: 182.52±4.8, mean ± SEM; Vesicle diameter:+shNS-EGFP: 34.28±0.84,+shC1ql2-EGFP: 35.37±0.21, mean ± SEM). Moreover, *C1ql2* KD did not affect the number of MFB, as quantified by the number of ZnT3$^+$ puncta in the SL of CA3 (*Figure 4—figure supplement 1h-i*;+shNS-EGFP: 1525.319±90.72,+shC1ql2-EGFP: 1547.94±48.51, mean ± SEM). To test whether shRNA-mediated KD of *C1ql2* expression also affects MF-LTP, we performed LTP recordings in *C1ql2* KD and control mice. Compared to controls, slices from *C1ql2* KD mice exhibited a significant reduction of LTP at 20–30 and 30–40 min time intervals, similarly to *Bcl11b* cKO animals (*Figure 4g–i*; 0–10 min:+shNS-EGFP: 105.0±4.0,+shC1ql2-EGFP: 94.3±4.5, 10–20 min:+shNS-EGFP: 56.3±4.5,+shC1ql2-EGFP: 35.1±2.8, 20–30 min:+shNS-EGFP: 50.2±4.5,+shC1ql2-EGFP: 23.4±3.5, 30–40 min:+shNS-EGFP: 44.6±4.3,+shC1ql2-EGFP: 20.1±4.1, mean ± SEM). *C1ql2* KD did not affect basal synaptic transmission, as evidenced by the respective input-output curves (*Figure 3—figure supplement 1c-d*;+shNS-EGFP: 2.50±0.16,+shC1ql2-EGFP: 2.45±0.14, mean ± SEM). Thus, KD of *C1ql2* in WT DGN recapitulates major phenotypes observed upon *Bcl11b* cKO, supporting that Bcl11b controls SV recruitment and LTP in hippocampal MFS specifically through C1ql2.

## C1ql2-Nrxn3(25b+) interaction recruits presynaptic vesicles in vitro and in vivo

C1ql2 was previously shown to interact with a particular splice variant of Nrxn3β containing exon 25b sequences, Nrxn3(25b+), which was recombinantly expressed in HEK293 cells (*Matsuda et al., 2016*). To explore whether the Bcl11b/C1ql2-dependent regulation of MFS involves interaction with neuronal Nrxn3, we co-cultured HEK293 cells that secreted myc-tagged C1ql2, to create regions of highly concentrated C1ql2, with primary hippocampal neurons transfected with GFP-Nrxn3α(25b). We used here the extracellularly longer Nrxn3α isoform because it is more strongly expressed in the murine DG compared to Nrxn3β (*Uchigashima et al., 2019*). C1ql2-dependent recruitment of Nrxn3α(25b+) was quantified by the surface area of HEK293 cells covered by GFP-Nrxn3α(25b+)-positive neuronal

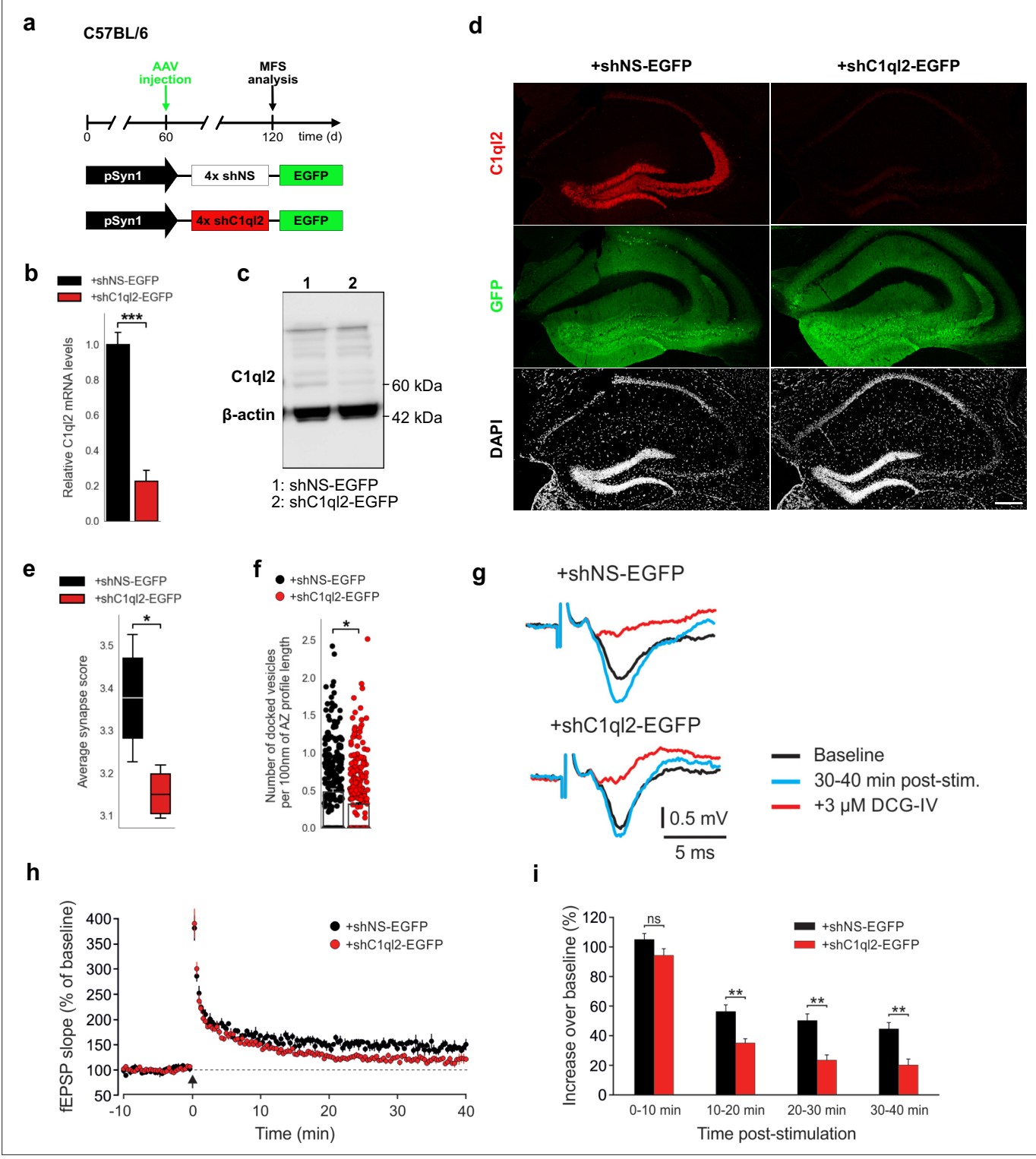

**Figure 4.** KD of *C1ql2* in DGN of WT mice impairs SV recruitment and LTP. (**a**) Experimental design to analyze the MFS after AAV-mediated KD of *C1ql2* in WT DGN. (**b**) Relative *C1ql2* mRNA levels in DGN. N=4. All data are presented as means; error bars indicate SEM. Unpaired t-test: ***p=0.0002. (**c**) Western blot of mouse hippocampal homogenates. (**d**) Immunohistochemistry of C1ql2 (red) and GFP (green) on hippocampal sections. Scale bar: 200 µm. (**e**) Average synapse score. N=4. Unpaired t-test. *p=0.025. (**f**) Number of docked vesicles per 100 nm AZ profile length. N=3. All data are presented as means; error bars indicate SEM. Points represent the individual examined AZ. Unpaired t-test. *p=0.018. (**g**) Representative fEPSP traces showing baselines before HFS (black), fEPSP changes 30–40 min after HFS (cyan) and following the application of 3 µM DCG-IV (red). (**h**) Time course of fEPSP slopes. The black arrow indicates HFS and the dashed line the baseline level. (**i**) Quantification of fEPSP facilitation at four different time intervals

*Figure 4 continued on next page*

*Figure 4 continued*

after HFS. Changes in fEPSP slope are shown as percentage of the mean baseline fEPSP. +shNS EGFP, 6 slices from 6 mice;+shC1ql2-EGFP, 7 slices from 7 mice. All data are presented as means; error bars indicate SEM. Mann-Whitney U-test for each time interval. 10–20 min: **p=0.0012; 20–30 min: **p=0.0023; 30–40 min: **p=0.0023; ns, not significant.

The online version of this article includes the following source data and figure supplement(s) for figure 4:

**Source data 1.** File containing the raw data for *Figure 4*, panels b, e-f & h and for *Figure 4—figure supplement 1*, panels b-g & i.

**Source data 2.** Original file for the western blot analysis in *Figure 4c*.

**Source data 3.** PDF containing *Figure 4c* and original scans of the relevant western blot analysis with highlighted bands and sample labels.

**Figure supplement 1.** *C1ql2* KD in DGN of WT mice impairs SV recruitment.

profiles (*Figure 5a*). HEK293 cells secreting C1ql2 had a significantly larger surface area covered by neuronal Nrxn3α(25b+) in comparison to HEK293 cells secreting myc-tag only (*Figure 5b*; myc-C1ql2: 39.97±3.99, myc-tag: 17.29±2.27, mean ± SEM). Using vGlut1 immunoreactivity as a proxy for SV localization (*Aoto et al., 2007*; *Fremeau et al., 2004*), we examined in the same system whether C1ql2-secreting HEK293 cells were able to cluster vGlut1 in surrounding GFP-Nrxn3α(25b+)-positive neurons (*Figure 5c*). Interestingly, vGlut1 accumulation was significantly increased in GFP-Nrxn3α(25b+)-positive neurons contacting C1ql2-secreting HEK293 cells as compared to C1ql2-negative HEK293 cells (*Figure 5d*; myc-C1ql2: 40.88±3.25, myc-tag: 24.78±4.99, mean ± SEM). To specifically analyze whether endogenous Nrxn3 is required for C1ql2-mediated vGlut1 accumulation in neurons, we co-cultured C1ql2-secreting HEK293 cells with primary hippocampal neurons derived from *Nrxn1, 2 & 3*<sup>flox/flox</sup> mice, in which all three *Nrxn* genes are floxed (*Chen et al., 2017*), and which we transfected with either active Cre recombinase or an inactive Cre (*Klatt et al., 2021*; *Figure 5e*). Neurons with the conditional triple *Nrxn* KO (*Nrxn* cTKO) showed a significantly lower accumulation of endogenous vGlut1 when contacting the C1ql2-secreting HEK293 cells compared to control neurons. Strikingly, selective reintroduction of the Nrxn3α(25b+) isoform into the *Nrxn* cTKO neurons was sufficient to normalize vGlut1 accumulation in vitro (*Figure 5f*; inactive Cre: 51.66±5.97, Cre: 27.83±2.83, Cre +Nrxn3α(25b+): 39.23±4.3, mean ± SEM). Collectively, our data strongly suggest that the C1ql2-mediated recruitment of vGlut1-positive SV in hippocampal neurons depends on the presence of Nrxn3α(25b+).

To explore the relevant epitope that mediates the binding of C1ql2 to Nrxn3(25b+) proteins, we analyzed the solvent accessible electrostatic surface properties of the C1ql-domain trimeric structure of C1ql2 (*Ressl et al., 2015*) (PDB_ID: 4QPY) and found that a change of lysine262 (K262) to glutamic acid renders a large area underneath the C1ql2-specific calcium and receptor binding loops negative (*Figure 5g*) and hypothesized that this would repel binding to Nrxn3(25b+). We generated a C1ql2.K262E variant, expressed it in HEK293 cells as before, and tested it for its ability to cluster Nrxn3α(25b+) as well as vGlut1 in contacting primary neurons (*Figure 5a–d*). In the presence of C1ql2.K262E, recruitment of Nrxn3α(25b+) was significantly lower compared to WT C1ql2 and indistinguishable from myc-tag control levels (*Figure 5a–b*; myc-K262E: 18.84±5.15). Moreover, the expression of C1ql2.K262E in HEK293 cells was unable to accumulate vGlut1 in contacting neurons expressing GFP-Nrxn3α(25b+) (*Figure 5c–d*; myc-K262E: 16.9±1.2, mean ± SEM). Together, these results provide in vitro evidence that the clustering of vGlut1 depends on an intact C1ql2-Nrxn3(25b+) interaction and that a single point mutation that creates a negative charge of that surface area underneath the C1ql2-specific calcium and receptor binding loops abolishes this binding activity and, thereby, the regulation of SV clustering.

To validate our identification of K262 as a key residue for the C1ql2-Nrxn3(25b+) interaction in vivo, we expressed C1ql2.K262E in *Bcl11b* cKO DGN, in which endogenous C1ql2 expression is downregulated by the ablation of *Bcl11b* (*Figure 6a*), while *Nrxn3* mRNA levels remain unaltered (*Figure 6—figure supplement 1a*; Control +EGFP: 1±0.173, Bcl11b cKO +EGFP: 1.14±0.27, mean ± SEM). AAV-mediated introduction of C1ql2.K262E in *Bcl11b* cKO DGN resulted in strong overall expression of the mutant protein (*Figure 6b–c*; Control +EGFP: 1±0.42, Bcl11b cKO +EGFP-2A-K262E: 9.68±4.75, mean ± SEM). However, the spatial distribution of C1ql2.K262E was notably different from the WT protein in the SL of CA3 where most of the MFS are located (*Figure 6d*). In the SL of CA3, protein levels of C1ql2.K262E were significantly lower compared to WT C1ql2 as quantified by the integrated fluorescence density (*Figure 6d–e*; Bcl11b cKO +EGFP-2A-C1ql2: 9.75±0.57 × 10<sup>4</sup>, Bcl11b

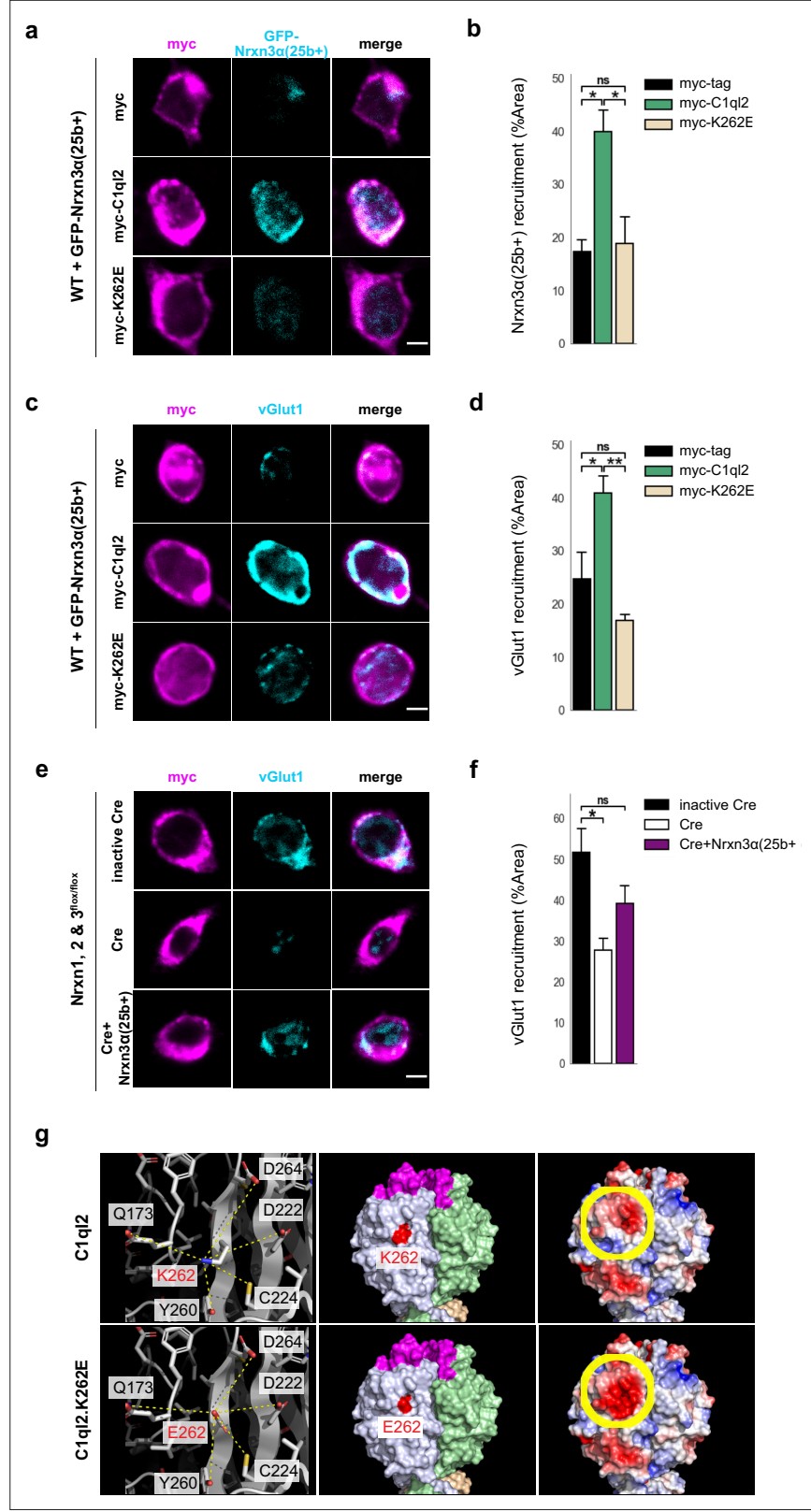

**Figure 5.** C1ql2-Nrxn3 interaction recruits vGlut1 in vitro. (**a**) Immunocytochemistry of myc-tagged C1ql2, C1ql2.
K262E or myc-tag (magenta) expressing HEK293 cells and GFP-Nrxn3α(25b+) (cyan) from contacting hippocampal
neurons. Scale bar: 5 μm. (**b**) Nrxn3α(25b+) recruitment by differentially transfected HEK293 cells. N=3. All data
are presented as means; error bars indicate SEM. One-way ANOVA and Tuckey's PHC. myc-C1ql2 vs. myc-tag:

*Figure 5 continued on next page*

*Figure 5 continued*

*p=0.016, and vs. myc-K262E: *p=0.022; ns, not significant. (**c**) Immunocytochemistry of myc-tagged C1ql2, C1ql2. K262E or myc-tag (magenta) expressing HEK293 cells and vGlut1 (cyan) from contacting hippocampal neurons. Scale bar: 5 µm. (**d**) vGlut1 recruitment by differentially transfected HEK293 cells. N=3. All data are presented as means; error bars indicate SEM. One-way ANOVA and Tuckey's PHC. myc-C1ql2 vs. myc-tag: *p=0.04, and vs. myc-K262E: **p=0.007; ns, not significant. (**e**) Immunocytochemistry of myc-tagged C1ql2 (magenta) expressing HEK293 cells and vGlut1 (cyan) from contacting control, *Nrxn123* KO or *Nrxn123* KO with Nrxn3α(25+) rescued hippocampal neurons. Scale bar: 5 µm. (**f**) vGlut1 recruitment by HEK293 cells in presence or absence of neuronal Nrxns. N=3. All data are presented as means; error bars indicate SEM. One-way ANOVA and Tuckey's PHC. inactive Cre vs. Cre: *p=0.023, and vs. Cre +Nrxn3α(25+): p=0.21; ns, not significant. (**g**) Trimeric structures of C1ql2 (PDB_ID: 4QPY, upper panels) and the variant C1ql2.K262E (lower panels). Residue 262 is the central residue (red, left and middle panels) of a larger area underneath the C1ql2-specific calcium and receptor binding loops (magenta, middle panel). The mutation K262E alters the charge of that surface area negative (yellow-circled area, right panels) and makes it potentially repulsive to bind Nrxn3(25b+).

The online version of this article includes the following source data for figure 5:

**Source data 1.** File containing the raw data for *Figure 5*, panels b, d, and f.

cKO +EGFP-2A-K262E: $5.89\pm0.55 \times 10^4$, mean ± SEM). The remaining signal of the C1ql2.K262E at the SL was equally distributed and in a punctate form, similar to WT C1ql2. As C1ql3 has been shown to form functional heteromers with C1ql2 at the MFS, we examined the spatial distribution of the C1ql3 protein upon AAV-mediated introduction of C1ql2.K262E for potential expression pattern changes but observed no overt difference (*Figure 6—figure supplement 1b*).

To investigate whether the remaining C1ql2.K262E affected the SV recruitment, we determined if C1ql2.K262E was able to recover the average synapse score in the *Bcl11b* cKO background, and found that the mutant C1ql2 variant did not rescue the SV distribution (*Figure 6f–g*, *Figure 6—figure supplement 1c*; Control +EGFP: 3.44±0.012, Bcl11b cKO +EGFP: 2.96±0.037, Bcl11b cKO +EGFP-2A-K262E: 2.87±0.043, mean ± SEM). Furthermore, the number of docked vesicles per 100 nm of AZ profile length in the MFB of animals receiving the C1ql2.K262E AAV was significantly lower compared to control animals and similar to that of *Bcl11b* mutants (*Figure 6h*, *Figure 6—figure supplement 1f-g*; Control +EGFP: 0.41±0.049, Bcl11b cKO +EGFP: 0.24±0.038, Bcl11b cKO +EGFP-2A-K262E: 0.19±0.025, mean ± SEM). The length of the AZ and the diameter of docked vesicles were unchanged (*Figure 6—figure supplement 1d-e*; AZ length: Control +EGFP: 167.94±7.35, Bcl11b cKO +EGFP: 161.9±5.56, Bcl11b cKO +EGFP-2A-K262E: 173.51±7.7, mean ± SEM; Vesicle diameter: Control +EGFP: 39.06±1.22, Bcl11b cKO +EGFP: 35.18±1.13, Bcl11b cKO +EGFP-2A-K262E: 36.38±2.19, mean ± SEM). Unexpectedly, however, C1ql2.K262E was able to rescue the loss of MF-LTP observed in *Bcl11b* cKO (*Figure 6—figure supplement 1h-j*; 0–10 min: Control +EGFP: 90.4±7.2, Bcl11b cKO +EGFP-2A-K262E: 155.8±30.3, 10–20 min: Control +EGFP: 42.7±3.6, Bcl11b cKO +EGFP-2A-K262E: 68.7±17.3, 20–30 min: Control +EGFP: 52.5±7.6, Bcl11b cKO +EGFP-2A-K262E: 55.9±13.8, 30–40 min: Control +EGFP: 50.1±7.3, Bcl11b cKO +EGFP-2A-K262E: 47.8±9.3, mean ± SEM). It has been shown that C1ql2 also interacts with specific postsynaptic KAR subunits (*Matsuda et al., 2016*). To test whether the C1ql2.K262E variant retained its ability to interact with GluK2, protein extract of HEK293 cells expressing either GluK2-myc-flag/GFP-C1ql2 or GluK2-myc-flag/GFP-C1ql2.K262E was examined by co-immunoprecipitation and revealed that both C1ql2 and C1ql2.K262E had GluK2 bound when precipitated (*Figure 6—figure supplement 1k*). Together, our data suggest that Bcl11b regulates MFS function through divergent C1ql2-dependent downstream signaling pathways: while SV recruitment depends on a direct interaction of C1ql2 with Nrxn3(25b+), C1ql2 appears to regulate MF-LTP through Nrxn3(25b+)-independent mechanisms.

To further explore whether binding to Nrxn3 is required for C1ql2-dependent regulation of SV recruitment, we stereotaxically injected an AAV expressing GFP-tagged Cre or inactive Cre into the DG of 2-month-old *Nrxn1, 2 & 3^flox/flox* mice (*Figure 7a*), which resulted in strong reduction of *Nrxn3* mRNA levels in DGN 2 months later. Only mild reduction of *Nrxn1* and unchanged expression of *Nrxn2* was observed (*Figure 7b*, *Figure 7—figure supplement 1a*; Nrxn1:+inactive Cre: 1±0.084,+Cre: 0.714±0.037; Nrxn2:+inactive Cre: 1±0.065,+Cre: 0.771±0.071; Nrxn3:+inactive Cre: 1±0.127,+Cre: 0.381±0.09, mean ± SEM). Fluorescence intensity of endogenous C1ql2 protein along the MF axons in the SL of CA3 was significantly reduced in *Nrxn* cTKO animals compared to *Nrxn1, 2 & 3^flox/flox*

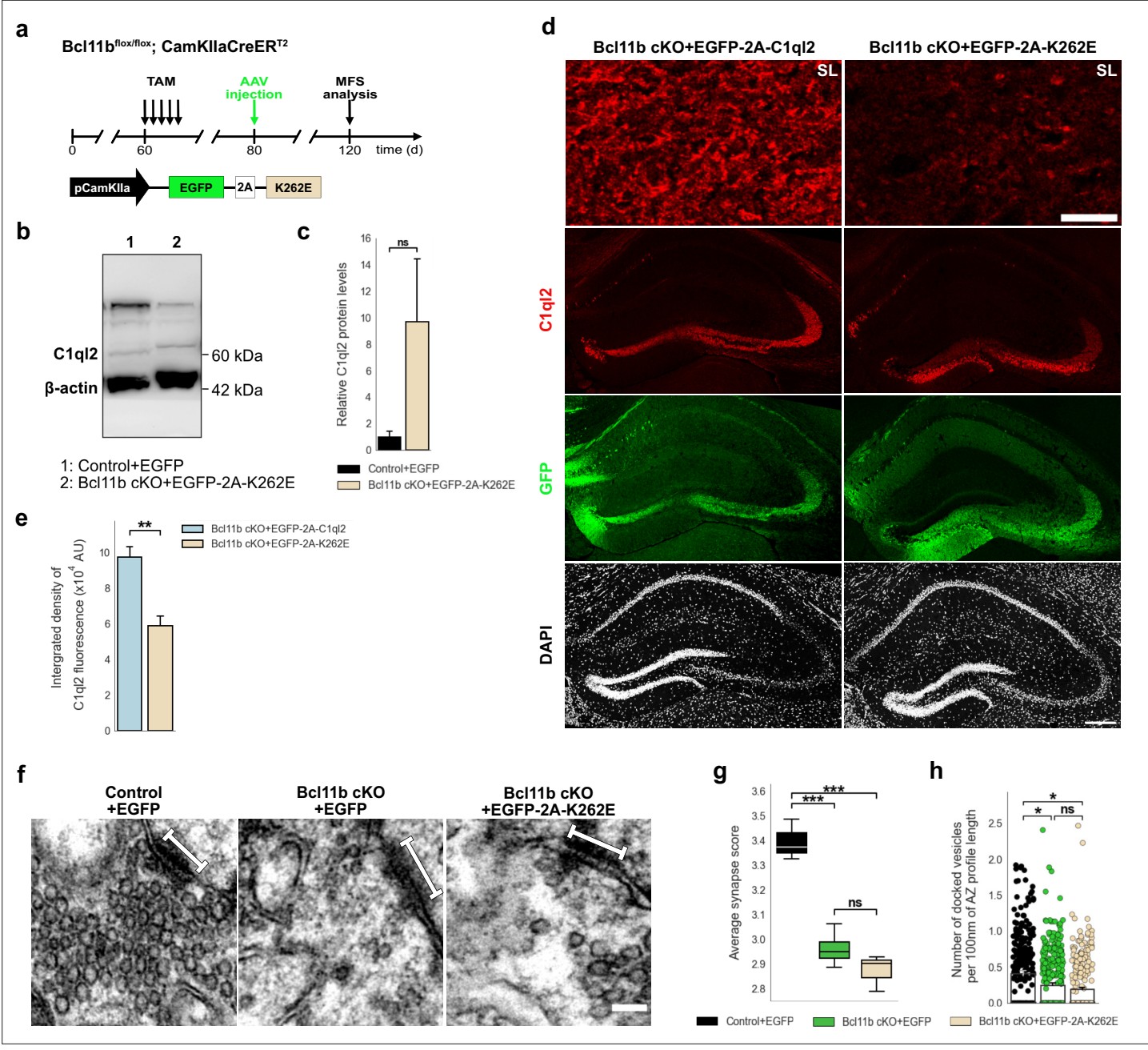

**Figure 6.** C1ql2-Nrxn3(25b+) interaction is important for C1ql2 localization at the MFS and SV recruitment. (**a**) Experimental design to analyze the MFS after AAV-mediated expression of C1ql2.K262E in *Bcl11b* cKO DGN. (**b**) Western blot and (**c**) relative C1ql2.K262E protein levels in mouse hippocampal homogenates. N=3. All data are presented as means; error bars indicate SEM. Mann-Whitney U-test. ns, not significant. (**d**) Immunohistochemistry of C1ql2 (red) and GFP (green) in hippocampal sections. Scale bar: 200 µm. Upper panels depict close-ups of C1ql2 staining from the SL of CA3. Scale bar: 15 µm. (**e**) Integrated density of C1ql2 fluorescence in the SL of CA3. N=3. All data are presented as means; error bars indicate SEM. Unpaired t-test. *p=0.008. (**f**) Electron microscope images of MFS and proximal SV. White bars mark synapse length from postsynaptic side. Scale bar: 100 nm. (**g**) Average synapse score. Control +EGFP, Bcl11b cKO +EGFP-2A-K262E: N=3; Bcl11b cKO +EGFP: N=4. Two-way ANOVA and Tuckey's PHC. Control +EGFP vs. Bcl11b cKO +EGFP: ***p=0.0004, and vs. Bcl11b cKO +EGFP-2A-K262E: ***p=0.0002; ns, not significant. (**h**) Number of docked vesicles per 100 nm AZ profile length. Control +EGFP, Bcl11b cKO +EGFP-2A-K262E: N=3; Bcl11b cKO +EGFP: N=4. All data are presented as means; error bars indicate SEM. Points represent the individual examined AZ. Two-way ANOVA and Tuckey's PHC. Control +EGFP vs. Bcl11b cKO +EGFP: *P=0.0434, and vs. Bcl11b cKO +EGFP-2A-K262E: *p=0.0196; ns, not significant. Data for Control +EGFP and Bcl11b cKO +EGFP-2A-K262E from f-h in this figure are compared with Bcl11b cKO +EGFP data from *Figure 2*.

The online version of this article includes the following source data and figure supplement(s) for figure 6:

**Source data 1.** File containing the raw data for *Figure 6*, panels c, e, and g-h and for *Figure 6—figure supplement 1*, panels a, c-g, and i.

*Figure 6 continued on next page*

*Figure 6 continued*

**Source data 2.** Original file for the western blot analysis in *Figure 6b*.

**Source data 3.** PDF containing *Figure 6b* and original scans of the relevant western blot analysis with highlighted bands and sample labels.

**Figure supplement 1.** C1ql2-Nrxn3(25b+) interaction is important for SV recruitment at the MFS.

**Figure supplement 1—source data 1.** Original file for the Western blot analysis in *Figure 6—figure supplement 1k*.

**Figure supplement 1—source data 2.** PDF containing *Figure 6—figure supplement 1k* and original scans of the relevant Western blot analysis with highlighted bands and sample labels.

animals expressing inactive Cre (*Figure 7c–d*;+inactive Cre: 11.72±1.63,+Cre: 4.71±0.93, mean ± SEM). However, *C1ql2* mRNA levels in DGN remained unchanged (*Figure 7—figure supplement 1b*;+inactive Cre: 1±0.23,+Cre: 0.8±0.11, mean ± SEM), suggesting that overall production of C1ql2 protein was not affected. To control for the specificity of this effect, we also determined the level of C1ql3 expression and found no overt changes in *Nrxn* cTKO (*Figure 7—figure supplement 1c*). To exclude that the reduced C1ql2 fluorescence intensity was simply a consequence of an overall loss of MFB, we used ZnT3 as a marker of MFB and found it unchanged in *Nrxn* cTKO compared to controls (*Figure 7e*). Remarkably, disruption of the C1ql2-Nrxn3(25b+) binding by ablation of *Nrxn3* in *Nrxn* cTKO mutants not only led to reduced C1ql2 fluorescence intensity (*Figure 7c–d*), but recapitulated the phenotype observed upon *Bcl11b* ablation or by KD of *C1ql2* as evidenced by a large reduction of the average synapse score in *Nrxn* cTKO (*Figure 7f–g*, *Figure 7—figure supplement 1d*;+inactive Cre: 3.11±0.06;+Cre: 2.67±0.074, mean ± SEM). Also, similarly to the *Bcl11b* and *C1ql2* mutant phenotypes, we observed the number of docked vesicles per 100 nm of AZ profile length in *Nrxn* cTKO to be diminished compared to controls (*Figure 7h*, *Figure 7—figure supplement 1e-f*;+inactive Cre: 0.404±0.035,+Cre: 0.195±0.02, mean ± SEM), whereas the AZ length and the diameter of docked vesicles remained unchanged (*Figure 7—figure supplement 1g-h*; AZ length:+inactive Cre: 193.2±6.88;+Cre: 188.44±11.43, mean ± SEM; Vesicle diameter:+inactive Cre: 38.38±0.44;+Cre: 37.12±0.8, mean ± SEM). Thus, our results provide evidence that Bcl11b controls MFS organization through C1ql2/Nrxn3(25b+)-dependent signaling, explicating how Bcl11b, a transcription factor with a broad range of functions, can regulate highly specific processes in the brain.

## Discussion

There is emerging evidence that the zinc finger transcription factor Bcl11b is involved in the pathogenesis of neurodevelopmental as well as neuropsychiatric disorders that are frequently associated with synaptic dysfunction. Previous work from our group demonstrated Bcl11b to be essential for synapse function in the mossy fiber circuit of the adult murine hippocampus. The underlying molecular mechanisms downstream of Bcl11b, however, remained elusive. In the present study, we uncover a novel C1ql2-dependent regulatory pathway through which Bcl11b controls the structural as well as functional integrity of hippocampal MFS in adult mice. We show that SV recruitment to the AZ of MFS, as well as the expression of MF-LTP, depend on C1ql2, which is a direct functional target of Bcl11b. Reintroduction of C1ql2 into *Bcl11b* mutant DGN restores defective SV recruitment and LTP expression. KD of *C1ql2* in DGN recapitulates the impaired SV recruitment and loss of LTP observed in *Bcl11b* mutants. Finally, we show that C1ql2 controls SV recruitment through a direct interaction with presynaptic Nrxn3(25b+), while LTP depends on C1ql2 signals independent of Nrxn3 interaction. Recent studies suggested Nrxn3, as well as C1ql2, to be associated with neuropsychiatric disorders (*Hishimoto et al., 2007*; *Hu et al., 2013*; *Huggett and Stallings, 2020a*; *Marballi et al., 2022*). Our study for the first time identifies a Bcl11b/C1ql2/Nrxn3-dependent signaling pathway in the control of basic structural and functional properties of MFS. Analysis of this regulatory pathway in mice may provide important novel insights into the pathogenesis of neurodevelopmental and neuropsychiatric disorders.

We have previously shown that conditional ablation of *Bcl11b* in the adult hippocampus leads to structural and functional changes of MFS characterized by an overall reduction in synapse numbers, loss of bouton complexity, misdistribution of SV as well as loss of MF-LTP (*De Bruyckere et al., 2018*). Here, we found that reintroduction of the synaptic organizer protein C1ql2, which is a direct transcriptional target of Bcl11b and is downregulated in *Bcl11b* mutant DGN (*De Bruyckere et al., 2018*), was

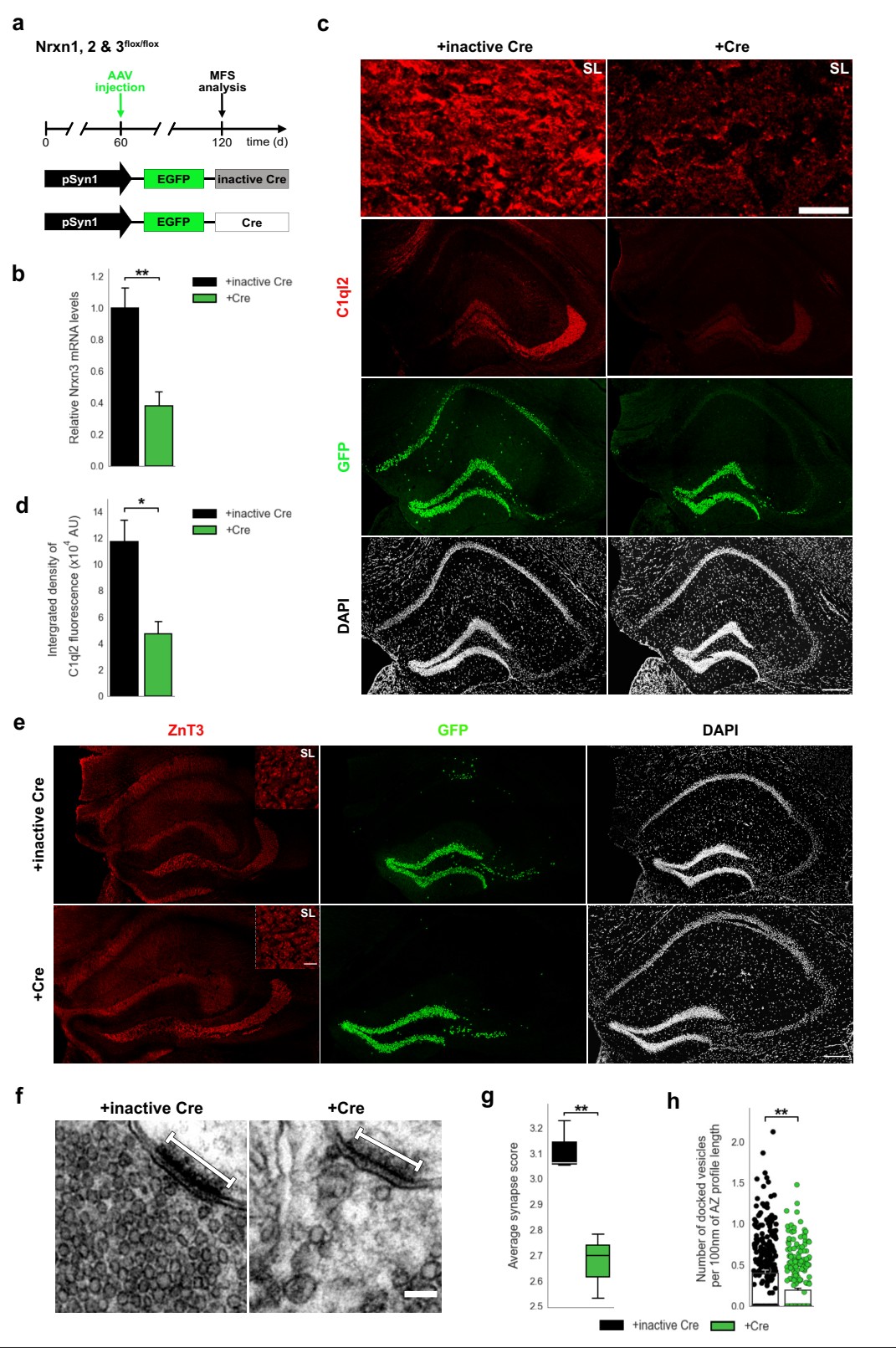

**Figure 7.** *Nrxn* KO perturbs C1ql2 localization at the MFS and SV recruitment. (**a**) Experimental design to analyze the MFS after AAV-mediated *Nrxn* KO. (**b**) Relative Nrxn3 mRNA levels. N=4. All data are presented as means; error bars indicate SEM. Unpaired t-test. **p=0.007. (**c**) Immunohistochemistry of C1ql2 (red) and GFP (green) in hippocampal sections. Scale bar: 200 µm. Upper panels depict close-ups of C1ql2 staining from the SL of CA3.

*Figure 7 continued on next page*

*Figure 7 continued*

Scale bar: 15 µm. (**d**) Integrated density of C1ql2 fluorescence in the SL of CA3. N=3. All data are presented as means; error bars indicate SEM. Unpaired t-test. *p=0.02. (**e**) Immunohistochemistry of ZnT3 (red) and GFP (green) in hippocampal sections. Scale bar: 200 µm. Upper right corner of ZnT3 panels depicts close-ups from the SL of CA3. Scale bar: 15 µm. (**f**) Electron microscope images of MFS and proximal SVs. White bars mark synapse length from postsynaptic side. Scale bar: 100 nm. (**g**) Average synapse score. N=3. Unpaired t-test. **p=0.009. (**h**) Number of docked vesicles per 100 nm AZ profile length. N=3. All data are presented as means; error bars indicate SEM. Points represent the individual examined AZ. Unpaired t-test. **p=0.007.

The online version of this article includes the following source data and figure supplement(s) for figure 7:

**Source data 1.** File containing the raw data for *Figure 7*, panels b, d, and g-h and for *Figure 7—figure supplement 1*, panels a-b and d-g.

**Figure supplement 1.** *Nrxn* KO perturbs SV recruitment at the MFS.

able to rescue major part of the *Bcl11b* mutant phenotype at the MFS. Restoring C1ql2 expression in *Bcl11b* cKO DGN led to a complete rescue of the SV distribution and docking, as well as LTP at the MFS, while synapse numbers and ultrastructural complexity of boutons remained unchanged. Furthermore, KD of *C1ql2* in WT DGN recapitulated the *Bcl11b* phenotype with impaired SV recruitment and loss of LTP, supporting the specificity of C1ql2 function. MF-LTP, which manifests as a long-term increase in presynaptic vesicle release probability ($P_r$) (*Shahoha et al., 2022*), directly depends on the distribution of SV in the proximity of the AZ. Recent studies have shown that the increase in $P_r$ involves the recruitment of new AZ and an increase in the number of docked and tethered vesicles, corresponding to the readily releasable pool of SV (*Orlando et al., 2021*; *Vandael et al., 2020*). The perturbed SV recruitment in both *Bcl11b* cKO and *C1ql2* KD mice could thus potentially explain the loss of LTP in both conditions. Indeed, the reintroduction of C1ql2 in *Bcl11b* cKO DGN specifically rescued SV recruitment and LTP, while synapse numbers and ultrastructural complexity of MFB remained unchanged. Whether the regulation of SV distribution by C1ql2 and the expression of MF-LTP are directly and causally linked remains to be determined. Additional factors have been suggested to contribute to the increase in $P_r$, including a tighter coupling between $Ca^{2+}$ channels and SV (*Midorikawa and Sakaba, 2017*) and the accumulation of $Ca^{2+}$ channels near release sites (*Fukaya et al., 2021*). It cannot, therefore, be excluded that C1ql2 regulates MF-LTP through one of these alternative mechanisms. Aiming to narrow in on the nature of the mechanism through which Bcl11b regulates MF-LTP, we used forskolin to induce LTP in *Bcl11b* cKO. MF-LTP relies on presynaptic mechanisms (*Castillo, 2012*; *Zalutsky and Nicoll, 1990*) and is mediated by the second messenger cAMP, which is produced by AC in response to $Ca^{2+}$ influx through voltage-gated $Ca^{2+}$ channels (*Li et al., 2007*) and KARs (*Lauri et al., 2001*; *Schmitz et al., 2003*). By using forskolin to directly activate AC, we bypassed these initial steps, and still found a reduction of LTP in *Bcl11b* cKO mice, similarly to HFS. These results strongly suggest that the loss of LTP is caused by a process downstream of the initial presynaptic $Ca^{2+}$ influx following stimulation. We note that, in the present experiments, we did not observe the decrease in input-output relation in *Bcl11b* cKO as reported in *De Bruyckere et al., 2018*. After excluding technical differences, e.g., different methods of data analysis, we conclude that the discrepancy is best explained by differences in the population of presynaptic fibers. In the present study, mossy fiber responses were specifically identified by testing for frequency facilitation and sensitivity to mGluR antagonists, whereas in the previous study, this purification was not done (*De Bruyckere et al., 2018*). It is not immediately obvious why the reduction in synapse numbers and misdistribution of SV in *Bcl11b* cKO animals does not affect basal synaptic transmission. While a modest displacement of SV might fail to noticeably influence synaptic transmission due to the low initial $P_r$ at MFS, causing only a fraction of release-ready vesicles to be initially released, the reduction in synapse numbers might indeed be expected to reflect in the input-output relationship. It might be that synapses that are preferentially eliminated in *Bcl11b* mutants are predominantly silent or have weak coupling strength, such that their loss has only a minimal effect on synaptic transmission. Further investigation is needed to elucidate this apparent discrepancy. Together, our results suggest Bcl11b to be an important synaptic regulator that controls the structure and function of adult MFS through both C1ql2-dependent, as well as -independent transcriptional programs.

C1ql proteins are complement-related factors that are synthesized by the presynapse and secreted into the synaptic cleft. Within the hippocampus, C1ql2 and –3 protein expression overlaps and is

highly restricted to DGN (*Iijima et al., 2010*) and the corresponding mossy fiber system, including MF-CA3 synapses. C1ql2 and –3 were previously suggested to form functional heteromers that can cluster postsynaptic KAR on MFS. Selective deletion of either C1ql2 or –3 in mice was reported to have no overt mutant hippocampal phenotype, suggesting functional compensation for both proteins (*Matsuda et al., 2016*). Using shRNA-mediated selective KD of *C1ql2* in DGN as well as rescue of the *Bcl11b* mutation by the reintroduction of C1ql2 into mutant DGN, we observed a novel, presynaptic function for C1ql2 in the recruitment of SV and the expression of LTP in MFS. This function was specific to C1ql2 since overexpression of C1ql3 in *Bcl11b* mutant DGN was unable to rescue the synapse phenotype. Furthermore, another study has identified all four C1ql proteins, including C1ql2, as ligands for the postsynaptic Brain-specific angiogenesis inhibitor 3 (BAI3). Addition of any of the four C1ql proteins to cultured hippocampal neurons led to a loss of excitatory synapses, a function inhibited by the presence of BAI3 (*Bolliger et al., 2011*). In our study, we show that neither loss of C1ql2 nor overexpression of C1ql2 affects the number of MFS, supporting the notion that synaptic organizers have synapse-specific functions. This highlights the role of C1ql2 as a synaptic organizer and adds a new layer of understanding to its function at the MFS.

Previous in vitro studies suggested that C1ql2 function at the MFS involves interaction with Nrxn3 isoforms containing the splice site 5 25b sequence (SS5[25b]) (*Matsuda et al., 2016*). Nrxns are synaptic cell adhesion molecules that mediate various synaptic properties (*Reissner et al., 2013*; *Südhof, 2017*), including the recruitment of SV and dense-core vesicles (*Dean et al., 2003*; *Ferdos et al., 2021*; *Quinn et al., 2017*; *Rui et al., 2017*). This prompted us to analyze, whether C1ql2-dependent SV recruitment in MFS requires a direct interaction with Nrxn3(25b+) in vitro and in vivo. Expression of C1ql2 in HEK293 cells co-cultured with GFP-Nrxn3α(25+)-expressing hippocampal neurons was able to recruit Nrxn3α(25b+) and vGlut1 at contact points, while C1ql2.K262E, a C1ql2 variant with an amino-acid replacement that perturbs the interaction with Nrxn3(25b+), was no longer able to recruit neuronal vGlut1. Furthermore, clustering of vGlut1 by C1ql2-secreting HEK293 cells was reduced in neurons harboring a pan-neurexin mutation, a phenotype that was rescued by the selective reintroduction of Nrxn3α(25b+). Finally, the introduction of C1ql2.K262E in *Bcl11b* cKO DGN in vivo was unable to rescue SV recruitment, while the silencing of *Nrxns* in DGN in vivo perturbed SV recruitment to a similar extent as in *Bcl11b* cKO and *C1ql2* KD. Based on these findings, we anticipated the overexpression of C1ql2.K262E in *Bcl11b* cKO DGN to be unable to rescue MF-LTP. Unexpectedly, the introduction of C1ql2.K262E into *Bcl11b* cKO fully rescued MF-LTP. This raises the possibility that C1ql2 can influence MF-LTP through additional, yet uncharacterized mechanisms, independent of SV recruitment or direct interaction with Nrxn3(25b+). We cannot exclude, however, that the expression of a mutant C1ql2 variant created an additional gain-of-function effect that circumvented SV recruitment and allowed the rescue of MF-LTP in our experimental system. The latter is supported by the fact that within the first 10 min after HFS, fEPSP slopes for C1ql2.K262E were significantly elevated compared to controls, an effect that was not seen after C1ql2 re-expression. Together, our data provide comprehensive experimental evidence that the direct interaction of C1ql2 with Nrxn3(25b+) is essential for SV recruitment at the MFS. Finally, we observed that an abolished interaction between C1ql2 and Nrxn3(25b+) was associated with reduced localization of the C1ql2 protein along the MF tract. This raises the possibility that C1ql2-Nrxn3 interaction might be involved in surface presentation of C1ql2, transportation, or stabilization at the MFS. The C1q domain can form stable, higher order oligomers (*Ressl et al., 2015*). Neurexins, on the other hand, are highly mobile outside and inside of synaptic terminals (*Klatt et al., 2021*; *Neupert et al., 2015*). Thus, the interaction of C1ql2 with Nrxn3(25b+) may reciprocally augment the accumulation of both proteins at synaptic sites.

*Neurexin* mRNAs are subjected to extensive alternative splicing that leads to the expression of thousands of isoforms with differential expression patterns (*Treutlein et al., 2014*; *Ullrich et al., 1995*) that act in a type-specific manner on synaptic functions (*Dai et al., 2019*; *Schreiner et al., 2014*; *Traunmüller et al., 2016*). Nrxn3 splice variants have been shown to regulate the function and plasticity of glutamatergic and GABAergic synapses through various mechanisms (*Aoto et al., 2013*; *Dai et al., 2019*; *Lloyd et al., 2023*; *Trotter et al., 2023*). The Nrxn3 splice site SS5 is a major contributor to the high number of Nrxn3 isoforms (*Schreiner et al., 2014*). One such isoform was recently found to be highly expressed in GABAergic interneurons at the DG, where it regulates dendritic inhibition (*Hauser et al., 2022*). Our findings on the role of Nrxn3 isoforms containing SS5[25b] in the recruitment

of SV at the MFS through interaction with C1ql2 add to the understanding of the synapse-specific mechanisms of action of Nrxns.

Perturbations in synaptic structure and function are major determinants of various neuropsychiatric and neurodevelopmental disorders (*Hayashi-Takagi, 2017*; *Lepeta et al., 2016*; *Zoghbi and Bear, 2012*). Emerging evidence from recent genetic studies suggests such disorders to be linked to various genes encoding for synaptic proteins (*Südhof, 2021*; *Torres et al., 2017*; *Wang et al., 2018*). Decoding the molecular mechanisms of synaptic organization and stability and their transcriptional regulation would therefore be expected to contribute to the mechanistic understanding of neuropsychiatric and neurodevelopmental disorders. The transcription factor Bcl11b has been linked to neurodevelopmental (*Lessel et al., 2018*), neurodegenerative (*Kunkle et al., 2016*; *Song et al., 2022*) and neuropsychiatric disorders (*Whitton et al., 2018*; *Whitton et al., 2016*). *BCL11B* mutations in humans are associated with neurodevelopmental delay, overall learning deficits as well as impaired speech acquisition and autistic features (*Eto et al., 2022*; *Lessel et al., 2018*; *Punwani et al., 2016*; *Yang et al., 2020*). Moreover, conditional ablation of *Bcl11b* selectively in the adult murine hippocampus results in impaired learning and memory behavior (*Simon et al., 2016*). *NRXN3* single-nucleotide polymorphisms (SNP) have been implicated in schizophrenia (*Hu et al., 2013*) and addiction (*Hishimoto et al., 2007*), with one recorded SNP located close to SS5 altering the expression of Nrxn3(25b+). Interestingly, recent studies have also associated *C1QL2* with schizophrenia (*Marballi et al., 2022*) as well as cocaine addiction (*Huggett and Stallings, 2020b*). In this study we demonstrate that Bcl11b, through its transcriptional target C1ql2, modulates the synaptic organization of MFS by controlling the recruitment of SV at AZ. This regulatory mechanism depends on a direct interaction of C1ql2 with Nrxn3(25b+). Importantly, SV trafficking and altered release probability have been implicated in neurological and neuropsychiatric disorders (*Egbujo et al., 2016*; *Lepeta et al., 2016*; *Zhu et al., 2021*). Thus, the identification of the Bcl11b/C1ql2/Nrxn3(25b+)-dependent signaling module in this study provides a new entry point for future mechanistic analyses of synaptopathies. Moreover, the existence of such cell-type-specific signaling modules reveals how a fundamental transcription factor with diverse functions such as Bcl11b can be implicated in the pathogenesis of brain disorders characterized by synaptic dysfunction.

# Materials and methods

## Animals

*Bcl11b* inducible mutants were generated as previously described (*De Bruyckere et al., 2018*). Bcl11b$^{flox/flox}$; CaMKIIa-CreER$^{T2}$ (*Bcl11b* cKO) and Bcl11b$^{+/+}$; CaMKIIa-CreER$^{T2}$ (control) littermates were used. The *Bcl11b* mutation was induced by intraperitoneal injection of 2 mg tamoxifen for five consecutive days. C57BL/6JRj mice were obtained from Janvier-Labs. For the pan-*neurexin* KO, *Nrxn1, 2 & 3$^{flox/flox}$* mice (*Chen et al., 2017*) were used. Animals were kept in a 12:12 hr light–dark cycle at a constant temperature (22 ± 1 °C) in IVC cages. All mouse experiments were carried out in compliance with the German law and approved by the respective government offices in Tübingen (TV Nr. 1224, Nr. 1517 and Nr. o.161–5) and Karlsruhe (TV Nr. 35–9185.81/G-310/19), Germany.

## Stereotaxic injections

For the expression of *C1ql2* and *C1ql3*, the DG of 80 days old Bcl11b cKO mice were injected with AAV vectors expressing EGFP-2A-C1ql2 and EGFP-2A-C1ql3, respectively. As control, *Bcl11b* cKO and control mice were injected with an AAV expressing EGFP. For the KD of C1ql2, the DG of 60-day-old C57BL/6JRj mice were injected with AAV 4 x shC1ql2-EGFP, expressing 4 shRNAs against *C1ql2* or control AAV 4xshNS-EGFP, expressing 4 x non-sense shRNAs. For pan-*neurexin* KO, the DG of 60 days old *Nrxn1, 2 and 3$^{flox/flox}$* mice were injected with an AAV expressing EGFP-Cre or a control AAV expressing EGFP-Cre.Y324F, an inactive Cre. All AAVs were produced by the Viral Vector Facility of the Neuroscience Center Zurich on request. The four selected non-sense shRNAs and the four shC1ql2 sequences were checked for and presented with no off-target bindings on the murine exome with up to two mismatches by siRNA-Check (http://projects.insilico.us/SpliceCenter/siRNACheck). The mice were anesthetized with 5% isoflurane and placed in a mouse stereotaxic apparatus. During the entire procedure, anesthesia was maintained by constant administration of 2.2% isoflurane. Eye ointment was applied to prevent eyes from drying. For electrophysiological experiments, mice were

subcutaneously injected with buprenorphine hydrochloride (0.1 mg/kg, Temgesic, Indivior) 30 min before and 3 hr after each surgery. For all other experiments, Butorphanol (Livisto) and Meloxicam (Boehringer-Ingelheim; 5 µg/g) were injected subcutaneously and the local anesthetic Bupivacaine (Puren; 5 µg/g) was injected subcutaneously at the incision site. After 10 min the head of the mouse was shaved and disinfected and an incision was made in the skin. Targeted injection sites were identified and a small craniotomy was performed for each site. The injector was placed at the individual sites and the viral solution was injected at 100 nL/min, with a 5–10 min recovery before removing the injector. After injections at all sites the incision was sutured and the animal was monitored for recovery from anesthesia, after which it was returned to its home cage. For histological and EM analyses of MFS, AAV were injected at three sites per hemisphere with the following coordinates (Bregma: AP 0; ML: 0; DV:0): AP –2 mm; ML ±1 mm; DV –2 mm. AP –2.5 mm; ML ±1.5 mm; DV –1.8 mm. AP –3.1 mm; ML ±2, DV –2.2 mm. For electrophysiological analyses, AAV were injected at two dorsoventral coordinates per hemisphere: AP –3.0 mm; ML ±3.25 mm; DV –2.4 and –2.8 mm. 200–300 nL of AAV (1e12 vg/mL) were injected in each location.

## RNA isolation and quantitative real-time PCR

All procedures were performed in an RNase-free environment. Animals were sacrificed under deep $CO_2$-induced anesthesia, brains were quickly dissected in ice-cold PBS, cryopreserved in 20% sucrose overnight, frozen in OCT compound (Polysciences), and stored at −80 °C. Twenty µm thick coronal sections were collected on UV-treated and 0.05% poly-L-lysine coated membrane-covered PEN slides (Zeiss), fixed for 1 min in ice-cold 70% EtOH, incubated for 45 sec in 1% cresyl violet acetate solution (Waldeck) and washed for 1 min each in 70% EtOH and 100% EtOH. Sections were briefly dried on a 37 °C warming plate and immediately processed. The granule cell layer of the DG was isolated by laser capture microdissection using a PALM MicroBeam Rel.4.2 (Zeiss). RNA was isolated from the collected tissue using Rneasy Micro Kit (Qiagen) and reverse transcribed using the SensiFast cDNA Synthesis Kit (Bioline). Quantitative real-time PCR was performed in triplets for each sample using the LightCycler DNA Master SYBR Green I Kit in a LightCycler 480 System (Roche). The relative copy number of Gapdh RNA was used for normalization. Data were analyzed using the comparative CT method (*Schmittgen and Livak, 2008*).

## Western blots

Briefly, hippocampi from freshly removed brains were dissected in ice-cold PBS, collected in Lysis Buffer (50 mM Tris pH 7.5, 150 mM NaCl, 0.5% sodium deoxycholate, 1% triton-X100, 0.1% SDS) and manually homogenized. Samples were centrifuged for 25 min at 13,200 rpm at 4 °C and the supernatant was collected. Protein concentration was calculated with Bradford assay. Protein suspension containing 40 µg of protein was mixed 1:1 with 2 x SDS loading dye (62.5 mM Tris, 10% Glycerol, 5% β-mercaptoethanol, 80 mM SDS, 1.5 mM bromophenol blue), boiled at 95 °C for 5 min, separated by SDS-PAGE and electrophoretically transferred onto PVDF membranes (Merck). Membranes were blocked with 5% non-fat milk (Sigma-Aldrich), incubated with mouse anti-β-actin (1:5000; Sigma-Aldrich) and rabbit anti-C1ql2 (1:500; Sigma-Aldrich), followed by Peroxidase-conjugated secondary antibodies (Jackson ImmunoResearch) and developed with Pierce ECL Western Blotting Substrate (Thermo Fisher Scientific). Signal was detected with ChemiDoc Imaging System (Bio-Rad) and analyzed with Image Lab Software (BioRad). Protein signal was normalized with the signal of β-actin.

## Immunohistochemistry and RNA in situ hybridization

Animals were sacrificed under deep $CO_2$-induced anesthesia, and brains were dissected in ice-cold 1 x PBS, and either fixed for 4 hr in 4% PFA in PBS at 4 °C and cryopreserved in 20% sucrose in PBS overnight at 4 °C or directly cryopreserved and then frozen in OCT compound (Polysciences). Sections were prepared at 14 µm. The unfixed sections were postfixed with 4% PFA in 1 x PBS for 20 min. Heat-induced antigen retrieval in 10 mM citrate buffer (pH 6.0) was performed for fixed sections. Sections were blocked at RT for 1 hr in 1 x PBS containing 0.1% TritonX-100 and 10% horse serum, and incubated overnight at 4 °C with primary antibodies, followed by a 90 min incubation with secondary antibodies. Sections were counterstained with DAPI (Molecular Probes). The following primary antibodies were used on fixed sections: guinea pig anti-Bcl11b (1:1000; *Simon et al., 2012*), rabbit anti-C1ql2 (1:1000; Invitrogen), rabbit anti-C1ql2 (1:500; Sigma-Aldrich), chicken anti-GFP (1:2000;

Abcam) and rabbit anti-C1ql3 (1:500; Biozol). Primary antibodies used on unfixed sections: mouse anti-vGlut1 (1:100; Synaptic Systems), guinea pig anti-Homer1 (1:250; Synaptic Systems), rabbit anti-C1ql2 (1:1000; Invitrogen) and rabbit anti-ZnT3 (1:200; Synaptic Systems). All fluorescent secondary antibodies were purchased from Jackson ImmunoResearch and used at 1:500 dilution. Hybridizations were performed with DIG-labelled riboprobes on 14-µm-thick sections.

## Transmission electron microscopy

Animals were sacrificed through $CO_2$-inhalation and immediately perfused transcardially with 0.9% NaCl for 1 min, followed by a fixative solution of 1.5% glutaraldehyde (Carl Roth) and 4% PFA in 0.1 M PB pH 7.2 for 13 min. Brains were dissected and postfixed in the fixative solution for 4 hr at 4 °C. Ultrathin sections (60 nm) were prepared and stained with lead citrate. Images were acquired using a transmission electron microscope LEO 906 (Zeiss) with a sharp-eye 2 k CCD camera and processed with ImageSP (Tröndle). Synapse score (*De Bruyckere et al., 2018*) was calculated according to the following criteria: 0–5 vesicles above the active zone = 0; 5–20 vesicles = 1; small group of vesicles ($\leq$200,000 $nm^2$) with distance between density and closest vesicle >100 nm=2; small group of vesicles ($\leq$200,000 $nm^2$) with distance between density and closest vesicle $\leq$100 nm=3; big group of vesicles (>200,000 $nm^2$) with distance between density and closest vesicle >100 nm=4; big group of vesicles (>200,000 $nm^2$) with distance between density and closest vesicle $\leq$100 nm=5. Synapses from approximately 30 MFB per animal were analyzed. Vesicles with a distance $\leq$5 nm from the plasma membrane were considered docked (*Kusick et al., 2022*; *Vandael et al., 2020*). Approximately 100 AZ per animal were analyzed.

## Electrophysiological recordings and data analysis

Animals were sacrificed under deep $CO_2$-induced anesthesia at 4 months. Brains were quickly removed and placed in ice-cold modified ACSF containing (in mM) 92 N-methyl-D-glucamine (NMDG), 2.5 KCl, 1.2 $NaH_2PO_4$, 30 $NaHCO_3$, 20 HEPES, 25 glucose, 5 Na-ascorbate, 2 thiourea, 3 Na-pyruvate, 10 $MgSO_4$, 0.5 $CaCl_2$, 6 N-acetyl-L-cysteine (NAC), saturated with carbogen gas (95% $O_2$ and 5% $CO_2$, pH 7.4) (*Ting et al., 2014*). 450-µm-thick horizontal slices were cut using a vibratome slicer (Leica) at a defined angle to improve the preservation of mossy fibers (*Bischofberger et al., 2006*). After cutting, slices were transferred to a 'Haas'-type interface chamber (*Haas et al., 1979*), where they were perfused with carbogen-saturated ACSF containing (in mM) 124 NaCl, 3 KCl, 2.3 $CaCl_2$, 1.8 $MgSO_4$, 10 glucose, 1.25 $NaH_2PO_4$, 26 $NaHCO_3$ (pH 7.4 at 34 °C) at a rate of 1.5 mL/min at 34 ± 1 °C. Slices were allowed to recover for a minimum of 1 hr before the start of recordings.

Recordings were carried out by placing a glass micropipette (tip diameter 3–5 µm) filled with ACSF in the SL of the CA3b area. To induce MF field excitatory post-synaptic potentials (fEPSP), a bipolar electrode (Science Products) was placed within the hilus region of the DG. 0.1 ms pulses were delivered with an Iso-Flex stimulus isolator (AMPI) at 20 s intervals. Putative mossy fiber signals were preliminarily identified using a 25 Hz train of five pulses. Input-output relationships were obtained by measuring the fiber volley amplitude and fEPSP slope in response to stimulations with intensities ranging from 3 to 40 V. For LTP recordings, stimulation intensity for each slice was adjusted to obtain a slope value of 20% (30% in the case of forskolin (Biomol) experiments) of the maximum fEPSP slope. LTP was induced by three trains of 100 stimulation pulses at 100 Hz (high-frequency stimulation, HFS), repeated every 8 s. 3 µM DCG-IV (Tocris Bioscience) was applied after each experiment, and only recordings displaying >70% reduction in putative MF-fEPSP slopes were used for analysis. fEPSPs were amplified 100 x with an EXT 10–2 F amplifier (npi electronics). Signals were low-pass filtered at 2 kHz and high-pass filtered at 0.3 Hz, digitized at 20 kHz with an analog-to-digital converter (Cambridge Electronic Design [CED]) and stored for offline analysis using Spike2 (v7) software (CED). Offline data analysis was performed on raw traces using Spike2. Slope values were measured from the linear part of the fEPSP rising phase by manually placing vertical cursors. Changes in fEPSP slopes were calculated as a percentage of the average baseline fEPSP ((average fEPSP slope in a given time interval after HFS – average fEPSP slope before HFS)/ (average fEPSP slope before HFS)).

## DNA constructs

For expression, *C1ql2* was cloned from mouse cDNA. A 6xHis-myc tag or GFP was attached to the N-terminus and the construct was cloned into the pSecTag2A vector (Invitrogen) in frame with

the N-terminal IgK signal sequence. A stop codon was introduced directly after *C1ql2*. The K262E point mutation was introduced with the Q5 Site-Directed Mutagenesis Kit (New England Biolabs). pSecTag2A was used for control experiments. Rat *Nrxn3α(25b+)* cDNA (*Ushkaryov and Südhof, 1993*) was inserted into an pSyn5 vector with human Synapsin promoter (*Neupert et al., 2015*) using BamHI and BglII. For the pan-*neurexin* KO and the control experiments, vectors with NLS-GFP-Cre or NLS-GFP-Cre.Y324F were used (*Klatt et al., 2021*; *Wang et al., 2016*). Expression vector for GluK2 was purchased from OriGene. All vectors were validated by sequencing (Eurofins Genomics).

## Primary hippocampal cultures

Hippocampi were dissected from P0 mice in HBSS media, digested for 15 min with HBSS containing 0.1% Trypsin (Gibco) at 37 °C, dissociated in plating media (MEM supplemented with 0.6% glucose, 10% FBS, 1% penicillin/streptomycin, DnaseI 4 U/mL) and seeded on poly-L-Lysin precoated coverslips placed inside 12-well plates at $1.5 \times 10^5$ cells/mL. After 3 hr, the plating media was replaced with neuronal growth media (Neurobasal A supplemented with 2% B27, 2 mM L-Glutamine, 1% penicillin/streptomycin, 1% N2 and 0.005% NGF). Cultures were kept at 37 °C under 5% $CO_2$ atmosphere. The day of plating was considered as 0 days in vitro (DIV). At DIV3 and DIV7 80% of the medium was exchanged with fresh growth medium. At DIV9 the medium was exchanged with penicillin/streptomycin-free growth medium and at DIV10 neurons were transfected using Lipofectamine 2000 (Invitrogen). Briefly, a total of 200 µL transfection mix per well was prepared by first mixing 100 µL Opti-MEM with 4 µL Lipofectamine 2000 in one tube and 100 µL Opti-MEM with 3 µg DNA in a different tube. After 5 min both volumes were combined and the mixture was incubated for 20 min at RT. The transfection mix was then added dropwise to the neurons. After 3 hr of incubation, the medium was exchanged with fresh growth medium.

## HEK293 cell culture

Human embryonic Kidney (HEK) 293 cells were obtained from ATCC and were maintained in DMEM supplemented with 10% fetal calf serum and 1% penicillin/streptomycin at 37 °C under 5% $CO_2$ atmosphere. Cells were transfected using Lipofectamine 2000 according to the manufacturer's instructions on the same day the neurons were transfected. Cells were incubated for at least 24 hr before being used in co-culture experiments.

## Neuronal and HEK293 co-culture and immunostaining

Transfected HEK293 cells were washed, dissociated, and resuspended in neuronal growth medium. $15 \times 10^3$ cells were added in each well containing DIV11 transfected neurons. HEK293 cells were co-cultured with the hippocampal primary neurons 2 days (DIV13 for neurons) before proceeding with immunostaining. Coverslips with cultured neurons and HEK293 cells were first fixed with 4% PFA in 1 x PBS for 10 min at 4 °C, then washed 3 x with 1 mL 1 x PBS and blocked with 1 x PBS containing 0.1% Triton X-100 and 10% horse serum for 1 hr at RT. Primary antibodies were incubated overnight at 4 °C, followed by a 90 min incubation with secondary antibodies. Cells were counterstained with DAPI (Molecular Probes). The following primary antibodies were used: rabbit anti-myc-tag (1:2000; Abcam), guinea pig anti-vGlut1 (1:250; Synaptic Systems), chicken anti-GFP (1:2000; Abcam). All fluorescent secondary antibodies were purchased from Jackson ImmunoResearch and used at 1:500 dilution. For each condition, 25 cells per experiment were analyzed.

## Structural protein modelling

The crystal structure of trimeric C1q-domains of mouse C1ql2 (*Ressl et al., 2015*) was used to predict a potential electrostatic binding site to splice insert 25 of Nrxn3α. An electrostatic surface map of the trimer was calculated using APBS (*Jurrus et al., 2018*). The K262E mutation was introduced using FoldX (foldxsuite.crg.eu) and was chosen in order to generate a negatively charged surface that would potentially be repulsive to Nrxn3α binding. Final models were visualized with PyMOL (https://pymol.org/2/).

## Co-immunoprecipitation

HEK293 cells were transfected using Lipofectamine 3000 according to the manufacturer's instructions and were incubated for at least 48 hr. Cells were harvested and proteins were extracted in lysis buffer

containing 25 mM Tris pH 7.4, 150 mM NaCl, 2 mM $MgCl_2$, 1% Igepal, 5% Glycerol, 1 x EDTA-free proteinase inhibitor, 0.5 mM DTT and 2.5 U/mL Benzonase. Protein A magnetic beads were washed 2 x with PBS including 0.02% Tween-20 and were incubated on a rotating wheel in RT for 2 hr with 2 µg of the following antibodies suspended in 200 µL 2 x with PBS/0.02% Tween-100: rabbit anti-IgG (Cell Signal), rabbit anti-flag (Sigma-Aldrich) or rabbit anti-C1ql2 (Sigma-Aldrich). Beads were washed 2 x with PBS/0.02% Tween-20, resuspended in 50 µL 2 x with PBS/0.02% Tween-20 and 40 µg protein extract was added. Beads were incubated o/n on a rotating wheel at 4 °C. Beads were thoroughly washed with 2 x with PBS/0.02% Tween-20, resuspended in 2 x SDS loading dye, and boiled for 10 min at 95 °C. Western blot (see above) was performed. Briefly, membranes were blocked with 5% non-fat milk (Sigma-Aldrich) and incubated with mouse anti-flag M2 (1:2000; Sigma-Aldrich). Three independent experiments were carried out.

### Image acquisition and analysis

All fluorescent images of sectioned hippocampal tissue were examined on a TCS SP5II confocal microscope (Leica) using LAS-X software and processed with Fiji (*Schindelin et al., 2012*). Overview images were acquired with a 20 x objective. Synapse numbers and ZnT3[+] puncta were quantified in the SL imaged with a 40 x objective at x2 zoom. C1ql2 fluorescence intensity was quantified in the SL imaged with a 40 x objective. Acquisition settings were kept constant for every sample and condition. All fluorescent images of co-cultured HEK293 cells were examined on a TCS SP8 confocal microscope (Leica) using LAS-X software and processed with Fiji. Images were acquired with a 40 x objective at 4 x zoom. As before, acquisition settings were kept constant for every sample and condition. Images were analyzed by masking transfected HEK293 cells and measuring the area of each mask covered by the chosen stain.

### Quantification and statistical analysis

Statistical analysis and graph generation was done using Python 3. If samples met the criteria for normality, we used two-tailed unpaired t-test to compare two groups and one-way ANOVA for more than two groups. For non-normally distributed data Mann-Whitney u-test was used. Two-way ANOVA was used for examining the influence of two different categorical independent variables. If ANOVAs were significant, we used a post hoc Tukey's multiple-comparisons test to compare groups (structural and expression data) or a post hoc Bonferonni's comparison test (electrophysiological data). Data are presented as mean ± SEM. Significance levels were set as indicated in figures: *p<0.05, **p<0.01, ***p<0.001.

## Acknowledgements

We thank L Schmid and J Andratschke (Ulm University) for their excellent technical support. We thank the staff of the core facility "Laser Microdissection" of the Medical Faculty of Ulm University. This work was supported by the Deutsche Forschungsgemeinschaft grants BR 2215/1–2 to SB, DR 326/13–2 to AD (239174087), and SFB 1348 TP A03 to MM. AK was partly supported by the international graduate school in molecular medicine, Ulm University. PL's work was supported by Innovation Technology Commission Funding (Health@InnoHK).

## Additional information

### Funding

| Funder | Grant reference number | Author |
|---|---|---|
| Deutsche Forschungsgemeinschaft | BR 2215/1-2 | Stefan Britsch |
| Deutsche Forschungsgemeinschaft | DR326/13-2 | Andreas Draguhn |
| Deutsche Forschungsgemeinschaft | SFB 1348 TP A03 | Markus Missler |

| Funder | Grant reference number | Author |
|---|---|---|
| International Graduate School in Molecular Medicine Ulm | | Artemis Koumoundourou |
| Innovation Technology Commission Funding | | Pengtao Liu |

The funders had no role in study design, data collection and interpretation, or the decision to submit the work for publication.

## Author contributions

Artemis Koumoundourou, Conceptualization, Formal analysis, Investigation, Writing – original draft, Writing – review and editing; Märt Rannap, Formal analysis, Investigation, Writing – original draft; Elodie De Bruyckere, Sigrun Nestel, Alexei V Egorov, Investigation; Carsten Reissner, Investigation, Methodology; Pengtao Liu, Resources; Markus Missler, Resources, Writing – original draft; Bernd Heimrich, Investigation, Methodology, Writing – original draft; Andreas Draguhn, Conceptualization, Supervision, Writing – original draft; Stefan Britsch, Conceptualization, Supervision, Funding acquisition, Writing – original draft, Writing – review and editing

## Author ORCIDs

Artemis Koumoundourou (iD) https://orcid.org/0000-0002-8917-5717
Elodie De Bruyckere (iD) http://orcid.org/0000-0001-6056-376X
Carsten Reissner (iD) http://orcid.org/0000-0002-5496-9971
Alexei V Egorov (iD) https://orcid.org/0000-0003-4899-8407
Markus Missler (iD) https://orcid.org/0000-0001-8008-984X
Stefan Britsch (iD) http://orcid.org/0000-0003-4379-3322

## Ethics

All mouse experiments were carried out in compliance with the German law and approved by the respective government offices in Tübingen (TV Nr. 1224, Nr. 1517 and Nr. o.161-5) and Karlsruhe (TV Nr. 35-9185.81/G-310/19), Germany.

Reviewer #1 (Public Review): https://doi.org/10.7554/eLife.89854.3.sa1
Reviewer #2 (Public Review): https://doi.org/10.7554/eLife.89854.3.sa2
Reviewer #3 (Public Review): https://doi.org/10.7554/eLife.89854.3.sa3
Author Response https://doi.org/10.7554/eLife.89854.3.sa4

# Additional files

## Supplementary files

• MDAR checklist

## Data availability

All data generated or analysed during this study are included in the manuscript and supporting files; source data files have been provided for all Figures.

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

# Appendix1

## Appendix 1—key resources table

| Reagent type (species) or resource | Designation | Source or reference | Identifiers | Additional information |
|---|---|---|---|---|
| Antibody | Anti-GFP (Chicken Polyclonal) | Abcam | Cat #Ab13970 RRID: AB_300798 | IHC(1:1000) IF(1:1000) |
| Antibody | Anti-Bcl11b (Guinea pig Polyclonal) | *Simon et al., 2012* | n/a | IHC(1:1000) |
| Antibody | Anti-Homer1 (Guinea pig Polyclonal) | Synaptic Systems | Cat #160004 RRID: AB_10549720 | IHC(1:250) |
| Antibody | Anti-vGlut1 (Mouse Monoclonal) | Synaptic Systems | Cat #135311 RRID: AB_887880 | IHC(1:100) |
| Antibody | Anti-flag M2 (Mouse Monoclonal) | Sigma-Aldrich | Cat #F3165 RRID: AB_259529 | WB(1:2000) |
| Antibody | Anti-vGlut1 (Guinea pig Polyclonal) | Synaptic Systems | Cat #135304 RRID: AB_887878 | ICC(1:250) |
| Antibody | Anti-β-actin (Mouse Monoclonal) | Sigma-Aldrich | Cat #A5441 RRID: AB_476744 | WB(1:5000) |
| Antibody | Anti-flag (Rabbit Polyclonal) | Sigma-Aldrich | Cat #F7425 RRID: AB_439687 | Co-IP(2 µg) |
| Antibody | Anti-IgG (Rabbit Isotype control) | Cell Signaling Technology | Cat #3900 RRID: AB_1550038 | Co-IP(2 µg) |
| Antibody | Ant-myc-tag (Rabbit Polyclonal) | Abcam | Cat #ab9106 RRID: AB_307014 | ICC(1:1000) |
| Antibody | Anti-C1ql2 (Rabbit Polyclonal) | Invitrogen | Cat #PA5-63504 RRID: AB_2638958 | IHC(1:1000) WB(1:500) |
| Antibody | Anti-C1ql2 (Rabbit Polyclonal) | Sigma-Aldrich | Cat #HPA057934 RRID: AB_2683558 | IHC(1:500) WB(1:500) |
| Antibody | Anti-C1ql3 (Rabbit Polyclonal) | Biozol | Cat #bs-9793R | IHC(1:500) |
| Antibody | Anti-ZnT3 (Rabbit Polyclonal) | Synaptic Systems | Cat #197003 RRID: AB_2737039 | IHC(1:2500) |
| Strain, strain background (adenovirus-associated virus) | AAV-DJ_8/2-hSyn1-chI[4xsh(mC1ql2)]-EGFP-WPRE-bGHp(A) | This paper | n/a | AAV expressing an shRNA cassette against mC1ql2 |
| Strain, strain background (adenovirus-associated virus) | AAV-DJ_8/2-hSynI-chI[4 x(m/rshNS)]-EGFP-WPRE-bGHp(A) | Viral Vector Facility, ZNZ | Cat #v668-DJ/8 | |
| Strain, strain background (adenovirus-associated virus) | AAV-DJ_8/2-mCaMKIIa-EGFP_2 A_C1QL2-WPRE-hGHp(A) | This paper | n/a | AAV expressing EGFP and C1ql2 |
| Strain, strain background (adenovirus-associated virus) | AAV-DJ_8/2-mCaMKIIa-EGFP_2 A_C1QL2.K262E-WPRE-hGHp(A) | This paper | n/a | AAV expressing EGFP and C1ql2 variant K262E |
| Strain, strain background (adenovirus-associated virus) | AAV-DJ_8/2-mCaMKIIa-EGFP_2 A_C1QL3-WPRE-hGHp(A) | This paper | n/a | AAV expressing EGFP and C1ql3 |
| Strain, strain background (adenovirus-associated virus) | AAV-DJ_8/2-hSyn1-chI-EGFP_Cre(Y324F)-WPRE-bGHp(A) | This paper | n/a | AAV expressing inactive Cre |
| Strain, strain background (adenovirus-associated virus) | AAV-DJ_8/2-hSyn1-chI-EGFP_iCre-WPRE-bGHp(A) | Viral Vector Facility, ZNZ | Cat #v750-DJ/8 | |

*Appendix 1 Continued on next page*

*Appendix 1 Continued*

| Reagent type (species) or resource | Designation | Source or reference | Identifiers | Additional information |
|---|---|---|---|---|
| Strain, strain background (adenovirus-associated virus) | AAV-DJ_8/2-mCaMKIIα-EGFP-WPRE-hGHp(A) | Viral Vector Facility, ZNZ | Cat #v113-DJ/8 | |
| Chemical compound, drug | B27 | Gibco | Cat #17504044 | |
| Chemical compound, drug | Benzonase | Millipore | Cat #71206 | |
| Chemical compound, drug | cOmplete EDTA-free proteinase inhibitor | Roche | Cat #11873580001 | |
| Chemical compound, drug | DCG-IV | Tocris Bioscience | Cat #0975 | |
| Chemical compound, drug | DMEM | Gibco | Cat #31966047 | |
| Chemical compound, drug | Fetal bovine serum | Gibco | Cat #10082147 | |
| Chemical compound, drug | Forskolin | Biomol | Cat #AG-CN2-0089 | |
| Chemical compound, drug | Igepal | Sigma-Aldrich | Cat #I3021 | |
| Chemical compound, drug | L-Glutamine | Gibco | Cat #25030149 | |
| Chemical compound, drug | N2 | Gibco | Cat #A1370701 | |
| Chemical compound, drug | Neurobasal A | Gibco | Cat #10888022 | |
| Chemical compound, drug | NGF | Gibco | Cat #13290010 | |
| Chemical compound, drug | Opti-MEM | Gibco | Cat #31985062 | |
| Chemical compound, drug | Poly-L-Lysine | Sigma-Aldrich | Cat #P2636 | |
| Chemical compound, drug | Tamoxifen | Sigma-Aldrich | Cat #T5648 | |
| Chemical compound, drug | Trypsin | Gibco | Cat #15090046 | |
| Commercial assay or kit | Dynabeads Protein A for Immunoprecipitation | Invitrogen | Cat #10001D | |
| Commercial assay or kit | LightCycler DNA Master SYBR Green I Master | Roche | Cat #04707516001 | |
| Commercial assay or kit | Lipofectamin 2000 | Invitrogen | Cat #11668030 | |
| Commercial assay or kit | Lipofectamin 3000 | Invitrogen | Cat #L3000001 | |
| Commercial assay or kit | Pierce ECL western blotting substrate | ThermoFisherScientific | Cat #32209 | |
| Commercial assay or kit | Q5 Site-Directed Mutagenesis Kit | New England Laboratories | Cat #E0554S | |
| Commercial assay or kit | RNeasy Micro Kit | Qiagen | Cat #74004 | |
| Cell line (Homo-sapiens) | Human Embryonic Kidney (HEK) 293 | ATCC | Cat #PTA-4488 RRID: CVCL_0045 | Female |
| Cell line (*M. musculus*) | Primary | This paper | n/a | Hippocampal primary neurons from P0 C57BL/6JRj mice |
| Cell line (*M. musculus*) | Primary | This paper | n/a | Hippocampal primary neurons from P0 *Nrxn1, 2 & 3* flox/flox mice |
| Strain, strain background (*M. musculus*) | Bcl11b flox/flox; CamKIIa-CreER T2 | *De Bruyckere et al., 2018* | n/a | |
| Strain, strain background (*M. musculus*) | C57BL/6JRj | Janvier Labs | RRID:MGI:2670020 | |
| Strain, strain background (*M. musculus*) | C57BL/6 N | Charles River Laboratories | Strain code: 027 | |

*Appendix 1 Continued*

| Reagent type (species) or resource | Designation | Source or reference | Identifiers | Additional information |
|---|---|---|---|---|
| Strain, strain background (*M. musculus*) | Nrxn*1, 2 & 3* <sup>flox/flox</sup> | *Jurrus et al., 2018* | n/a | |
| Recombinant DNA reagent | pAAV-8/2-hSyn1-chI[4xsh(mC1ql2)]-EGFP-WPRE-bGHp(A) | This paper | n/a | Plasmid for production of relevant AAV |
| Recombinant DNA reagent | pAAV-DJ_8/–2-mCaMKIIa-EGFP_2 A_C1QL2-WPRE-hGHp(A) | This paper | n/a | Plasmid for production of relevant AAV |
| Recombinant DNA reagent | pAAV-DJ_8/–2-mCaMKIIa-EGFP_2 A_C1QL2.K262E-WPRE-hGHp(A) | This paper | n/a | Plasmid for production of relevant AAV |
| Recombinant DNA reagent | pAAV-DJ_8/–2-mCaMKIIa-EGFP_2 A_C1QL3-WPRE-hGHp(A) | This paper | n/a | Plasmid for production of relevant AAV |
| Recombinant DNA reagent | pAAV-DJ_8/2-hSyn1-chI-EGFP_Cre(Y324F)-WPRE-bGHp(A) | This paper | n/a | Plasmid for production of relevant AAV |
| Recombinant DNA reagent | pCMV-GluK2-myc-flag (plasmid) | OriGene | Cat #MR219233 | |
| Recombinant DNA reagent | pCMV-Igk-GFP-C1ql2 | This paper | n/a | Expression plasmid for secreted GFP tagged C1ql2 |
| Recombinant DNA reagent | pCMV-Igk-GFP-C1ql2.K262E | This paper | n/a | Expression plasmid for secreted GFP tagged C1ql2 variant K262E |
| Recombinant DNA reagent | pCMV-Igk-His-myc-C1ql2 | This paper | n/a | Expression plasmid for secreted His-myc tagged C1ql2 |
| Recombinant DNA reagent | pCMV-Igk-His-myc-C1ql2.K262E | This paper | n/a | Expression plasmid for secreted His-myc tagged C1ql2 variant K262E |
| Recombinant DNA reagent | pSecTag2A | Invitrogen | Cat #V90020 | |
| Recombinant DNA reagent | pSyn1-EGFP-Nrxn3a(25b+) | This paper | n/a | Expression plasmid for GFP tagged Nrxn3a(25b+) |
| Recombinant DNA reagent | pSyn1-nls-EGFP-Cre (plasmid) | *Wang et al., 2016* | n/a | |
| Recombinant DNA reagent | pSyn1-nls-EGFP-Cre.Y324F (plasmid) | *Klatt et al., 2021* | n/a | |
| Software, algorithm | Fiji v2.14.0 | NIH | RRID:SCR_002285 | |
| Software, algorithm | CorelDRAW | Corel Corporation | X4 | |
| Software, algorithm | GraphPad v3.10 | InStat | RRID:SCR_000306 | |
| Software, algorithm | ImageSP | Tröndle | n/a | |
| Software, algorithm | Leica Application Suite X | Leica | RRID:SCR_013673 | |
| Software, algorithm | SigmaPlot v11.0 | Systat | RRID:SCR_003210 | |
| Software, algorithm | Spike2 v7 | CED | RRID:SCR_000903 | |

