## [Editor Report · eLife assessment]

The authors identify a new role for C1ql2 at mossy fiber synapses in the hippocampus and **convincingly** find that C1ql2, whose expression is controlled by Bcl11b, controls the recruitment of synaptic vesicles to active zones and is necessary for synaptic plasticity. These **important** results build upon prior discoveries of how Bcl11b, a disease-relevant molecule, contributes to our understanding of mossy-fiber synaptic development.

---

## [Referee Report · Reviewer #1 (Public Review)]

Koumoundourou et al., identify a pathway downstream of Bcl11b that controls synapse morphology and plasticity of hippocampal mossy fiber synapses. Using an elegant combination of in vivo, ex vivo, and in vitro approaches, the authors build on their previous work that indicated C1ql2 as a functional target of Bcl11b (De Bruyckere et al., 2018). Here, they examine the functional implications of C1ql2 at MF synapses in Bcl11b cKO mice and following C1ql2 shRNA. The authors find that Bcl11b KO and shRNA against C1ql2 significantly reduces the recruitment of synaptic vesicles and impairs LTP at MF synapses. Importantly, the authors test a role for the previously identified C1ql2 binding partner, exon 25b-containing Nrxn3 (Matsuda et al., 2016), as relevant at MF synapses to maintain synaptic vesicle recruitment. To test this, the authors developed a K262E C1ql2 mutant that disrupts binding to Nrxn3. Curiously, while Bcl11b KO and C1ql2 KD largely phenocopy (reduced vesicle recruitment and impaired LTP), only vesicle recruitment is dependent on C1ql2-Nrxn3 interactions. These findings provide new insight into the functional role of C1ql2 at MF synapses. The authors utilize a multidisciplinary approach to convincingly demonstrate a role for C1ql2-Nrxn3(25b+) interactions for vesicle recruitment and a Nrxn3(25b+)-independent role for C1ql2 in LTP, The authors establish an important signaling pathway that offers insight into how disruptions of Bcl11b contribute to synapse dysfunction and provide a much needed advance toward understanding the functional consequences of neurexin alternative splicing.

---

## [Referee Report · Reviewer #2 (Public Review)]

This manuscript describes experiments that further investigate the actions of the transcription factor Bcl11b in regulating mossy fiber (MF) synapses in the hippocampus. Prior work from the same group had demonstrated that loss of Bcl11b results in loss of MF synapses as well as a decrease in LTP. Here the authors focus on a target of Bcl11b a secreted synaptic organizer C1ql2 which is almost completed lost in Bcl11b KO. Viral reintroduction of C1ql2 rescues the synaptic phenotypes, whereas direct KD of C1ql2 recapitulates the Bcl1 phenotype. C1ql2 itself interacts directly with Nrxn3 and replacement with a binding deficient mutant C1q was not able to rescue the Bcl11b KO phenotype. Overall there are some interesting observations in the study, however there are also some concerns about the measures and interpretation of data.

The authors state they used a differential transcriptomic analysis to screen for candidate targets of Bcl11b, yet they do not present any details of this screen. This should be included and at the very least a table of all DE genes included. It is likely that many other genes are also regulated by Bcl11b so it would be important to the reader to see the rationale for focusing attention on C1ql2 in this study.

All viral mediated expression uses AAVs which are known to ablate neurogenesis in the DG (Johnston DOI: 10.7554/eLife.59291) through the ITR regions and leads to hyperexcitability of the dentate. While it is not clear how this would impact the measurements the authors make in MF-CA3 synapses, this should be acknowledged as a potential caveat in this study.

The authors claim that the viral re-introduction "restored C1ql2 protein expression to control levels. This is misleading given that the mean of the data is 2.5x the control (Figure 1d and also see Figure 6c). The low n and large variance are a problem for these data. Moreover, they are marked ns but the authors should report p values for these. At the least this likely large overexpression and variability should be acknowledged. In addition, the use of clipped bands on Western blots should be avoided. Please show the complete protein gel in primary figures of supplemental information.

Measurement of EM micrographs: As prior work suggested that MF synapse structure is disrupted the authors should report active zone length as this may itself affect "synapse score" defined by the number of vesicles docked. More concerning is that the example KO micrographs seem to have lost all the densely clustered synaptic vesicles that are away from the AZ in normal MF synapses e.g. compare control and KO terminals in Fig 2a or 6f or 7f. These terminals look aberrant and suggest that the important measure is not what is docked but what is present in the terminal cytoplasm that normally makes up the reserve pool. This needs to be addressed with further analysis and modifications to the manuscript.

The study also presents correlated changes in MF LTP in Bcl11b KO which are rescued by C1ql2 expression. It is not clear whether the structural and functional deficits are causally linked and this should be made clearer in the manuscript. It is also not apparent why this functional measure was chosen as it is unlikely that C1ql2 plays a direct role in presynaptic plasticity mechanisms that are through a cAMP/ PKA pathway and likely disrupted LTP is due to dysfunctional synapses rather than a specific LTP effect. The authors should consider measures that might support the role of Bcl11b targets in SV recruitment during depletion of synapses or measurements of the readily releasable pool size that would complement their finding in structural studies.

Bcl11b KO reduces the number of synapses, yet the I-O curve reported in Supp Fig 2 is not changed. How is that possible? This should be explained.

Matsuda et al DOI: 10.1016/j.neuron.2016.04.001 previously reported that C1ql2 organizes MF synapses by aligning postsynaptic kainate receptors with presynaptic elements. As this may have consequences for the functional properties of MF synapses including their plasticity, the authors should report whether they see deficient postsynaptic glutamate receptor signaling in the Bcl11b KO and rescue in the C1ql2 re-expression.

These are all addressed in the revised version.

---

## [Referee Report · Reviewer #3 (Public Review)]

Overall, this is a strong manuscript that uses multiple current techniques to provide specific mechanistic insight into prior discoveries of the contributions of the Bcl11b transcription factor to mossy fiber synapses of dentate gyrus granule cells. The authors employ an adult deletion of Bcl11b via Tamoxifen-inducible Cre and use immunohistochemical, electron microscopy, and electrophysiological studies of synaptic plasticity, together with viral rescue of C1ql2, a direct transcriptional target of Bcl11b or Nrxn3, to construct a molecular cascade downstream of Bcl11b for DG mossy fiber synapse development. They find that C1ql2 re-expression in Bcl11b cKOs can rescue the synaptic vesicle docking phenotype and the impairments in MF-LTP of these mutants. They also show that C1ql2 knockdown in DG neurons can phenocopy the vesicle docking and plasticity phenotypes of the Bcl11b cKO. They also use artificial synapse formation assays to suggest that C1ql2 functions together with a specific Nrxn3 splice isoform in mediating MF axon development, extending these data with a C1ql2-K262E mutant that purports to specifically disrupt interactions with Nrxn3. All of the molecules involved in this cascade are disease-associated and this study provides an excellent blueprint for uncovering downstream mediators of transcription factor disruption. Together this makes this work of great interest to the field. Strengths are the sophisticated use of viral replacement and multi-level phenotypic analysis while weaknesses include the linkage of C1ql2 with a specific Nrxn3 splice variant in mediating these effects.

Here is an appraisal of the main claims and conclusions:

1. C1ql2 is a downstream target of Bcl11b which mediates the synaptic vesicle recruitment and synaptic plasticity phenotypes seen in these cKOs. This is supported by the clear rescue phenotypes of synapse anatomy (Fig.2) and MF synaptic plasticity (Fig.3). One weakness here is the absence of a control assessing over-expression phenotypes of C1ql2. It's clear from Fig.1D that viral rescue is often greater than WT expression (totally expected). In the case where you are trying to suppress a LoF phenotype, it is important to make sure that enhanced expression of C1ql2 in a WT background does not cause your rescue phenotype. A strong overexpression phenotype in WT would weaken the claim that C1ql2 is the main mediator of the Bcl11b phenotype for MF synapse phenotypes.

2. Knockdown of C1ql2 via 4 shRNAs is sufficient to produce the synaptic vesicle recruitment and MF-LTP phenotypes. This is supported by clear effects in the shRNA-C1ql2 groups as compared to nonsense-EGFP controls. One concern (particularly given the use of 4 distinct shRNAs) is the potential for off-target effects, which is best controlled for by a rescue experiment with RNA-insensitive C1ql2 cDNA as opposed to nonsense sequences, which may not elicit the same off-target effects.

3. C1ql2 interacts with Nrxn3(25b+) to facilitate MF terminal SV clustering. This claim is theoretically supported by the HEK cell artificial synapse formation assay (Fig.5), the inability of the K262-C1ql2 mutation to rescue the Bcl11b phenotype (Fig.6) and the altered localization of C1ql2 in the Nrxn1-3 deletion mice (Fig.7). Each of these lines of experimental evidence has caveats that should be acknowledged and addressed. Given the hypothesis that C1ql2 and Nrxn3b(25b) are expressed in DG neurons and work together, the heterologous co-culture experiment seems weird. Up till now, the authors are looking at pre-synaptic function of C1ql2 since they are re-expressing it in DGNs. The phenotypes they are seeing are also pre-synaptic and/or consistent with pre-synaptic dysfunction. In Fig.5, they are testing whether C1ql2 can induce pre-synaptic differentiation in trans, i.e. theoretically being released from the 293 cells "post-synaptically". But the post-synaptic ligands (Nlgn1 and GluKs) are not present in the 293 cells, so a heterologous synapse assay doesn't really make sense here. The effect that the authors are seeing likely reflects the fact that C1ql2 and Nrxn3 do bind to each other, so C1ql2 is acting as an artificial post-synaptic ligand, in that it can cluster Nrxn3 which in turn clusters synaptic vesicles. But this does not test the model that the authors propose (i.e. C1ql2 and Nrxn3 are both expressed in MF terminals). Perhaps a heterologous assay where GluK2 is put into HEK cells and the C1ql2 and Nrxn3 are simultaneously or individually manipulated in DG neurons?

4. K262-C1ql2 mutation blocks the normal rescue through a Nrxn3(25b) mechanism (Fig.6). The strength of this experiment rests upon the specificity of this mutation for disrupting Nrxn3b binding (presynaptic) as opposed to any of the known postsynaptic C1ql2 ligands such as GluK2. While this is not relevant for interpreting the heterologous assay (Fig.5), it is relevant for the in vivo phenotypes in Fig.6. Similar approaches as employed in this paper can test whether binding to other known postsynaptic targets is altered by this point mutation.

5. Altered localization of C1ql2 in Nrxn1-3 cKOs. These data are presented to suggest that Nrx3(25b) is important for localizing C1ql2 to the SL of CA3. Weaknesses of this data include both the lack of Nrxn specificity in the triple a/b KOs as well as the profound effects of Nrxn LoF on the total levels of C1ql2 protein. Some measure that isn't biased by this large difference in C1ql2 levels should be attempted (something like in Fig.1F).

---

## [Author Response]

The following is the authors’ response to the original reviews.

**Public Reviews:**

**Reviewer #1 (Public Review):**
Koumoundourou et al., identify a pathway downstream of Bcl11b that controls synapse morphology and plasticity of hippocampal mossy fiber synapses. Using an elegant combination of in vivo, ex vivo, and in vitro approaches, the authors build on their previous work that indicated C1ql2 as a functional target of Bcl11b (De Bruyckere et al., 2018). Here, they examine the functional implications of C1ql2 at MF synapses in Bcl11b cKO mice and following C1ql2 shRNA. The authors find that Bcl11b KO and shRNA against C1ql2 significantly reduces the recruitment of synaptic vesicles and impairs LTP at MF synapses. Importantly, the authors test a role for the previously identified C1ql2 binding partner, exon 25b-containing Nrxn3 (Matsuda et al., 2016), as relevant at MF synapses to maintain synaptic vesicle recruitment. To test this, the authors developed a K262E C1ql2 mutant that disrupts binding to Nrxn3. Curiously, while Bcl11b KO and C1ql2 KD largely phenocopy (reduced vesicle recruitment and impaired LTP), only vesicle recruitment is dependent on C1ql2-Nrxn3 interactions. These findings provide new insight into the functional role of C1ql2 at MF synapses. While the authors convincingly demonstrate a role for C1ql2-Nrxn3(25b+) interaction for vesicle recruitment and a Nrxn3(25b+)independent role for C1ql2 in LTP, the underlying mechanisms remain inconclusive. Additionally, a discussion of how these findings relate to previous work on C1ql2 at mossy fiber synapses and how the findings contribute to the biology of Nrxn3 would increase the interpretability of this work.

As suggested by reviewer #1, we extended our discussion of previous work on C1ql2 and additionally discussed the biology of Nrxn3 and how our work relates to it. Moreover, we extended our mechanistic analysis of how Bcl11b/C1ql2/Nrxn3 pathway controls synaptic vesicle recruitment as well as LTP (please see also response to reviewer #2 points 5 and 8 and reviewer #3 point 4 of public reviews below for detailed discussion).

**Reviewer #2 (Public Review):**
This manuscript describes experiments that further investigate the actions of the transcription factor Bcl11b in regulating mossy fiber (MF) synapses in the hippocampus. Prior work from the same group had demonstrated that loss of Bcl11b results in loss of MF synapses as well as a decrease in LTP. Here the authors focus on a target of Bcl11b a secreted synaptic organizer C1ql2 which is almost completely lost in Bcl11b KO. Viral reintroduction of C1ql2 rescues the synaptic phenotypes, whereas direct KD of C1ql2 recapitulates the Bcl1 phenotype. C1ql2 itself interacts directly with Nrxn3 and replacement with a binding deficient mutant C1q was not able to rescue the Bcl11b KO phenotype. Overall there are some interesting observations in the study, however there are also some concerns about the measures and interpretation of data.The authors state that they used a differential transcriptomic analysis to screen for candidate targets of Bcl11b, yet they do not present any details of this screen. This should be included and at the very least a table of all DE genes included. It is likely that many other genes are also regulated by Bcl11b so it would be important to the reader to see the rationale for focusing attention on C1ql2 in this study.

The transcriptome analysis mentioned in our manuscript was published in detail in our previous study (De Bruyckere et al., 2018), including chromatin-immunoprecipitation that revealed C1ql2 as a direct transcriptional target of Bcl11b. Upon revision of the manuscript, we made sure that this was clearly stated within the main text module to avoid future confusion. In the same publication (De Bruyckere et al., 2018), we discuss in detail several identified candidate genes such as Sema5b, Ptgs2, Pdyn and Penk as putative effectors of Bcl11b in the structural and functional integrity of MFS. C1ql2 has been previously demonstrated to be almost exclusively expressed in DG neurons and localized to the MFS.

There it bridges the pre- and post-synaptic sides through interaction with Nrxn3 and KAR subunits, respectively, and regulates synaptic function (Matsuda et al., 2016). Taken together, C1ql2 was a very good candidate to study as a potential effector downstream of Bcl11b in the maintenance of MFS structure and function. However, as our data reveal, not all Bcl11b mutant phenotypes were rescued by C1ql2 (see supplementary figures 2d-f of revised manuscript). We expect additional candidate genes, identified in our transcriptomic screen, to act downstream of Bcl11b in the control of MFS.

All viral-mediated expression uses AAVs which are known to ablate neurogenesis in the DG (Johnston DOI: 10.7554/eLife.59291) through the ITR regions and leads to hyperexcitability of the dentate. While it is not clear how this would impact the measurements the authors make in MF-CA3 synapses, this should be acknowledged as a potential caveat in this study.

We agree with reviewer #2 and are aware that it has been demonstrated that AAV-mediated gene expression ablates neurogenesis in the DG. To avoid potential interference of the AAVs with the interpretability of our phenotypes, we made sure during the design of the study that all of our control groups were treated in the same way as our groups of interest, and were, thus, injected with control AAVs. Moreover, the observed phenotypes were first described in Bcl11b mutants that were not injected with AVVs (De Bruyckere et al., 2018). Finally, we thoroughly examined the individual components of the proposed mechanism (rescue of C1ql2 expression, over-expression of C1ql3 and introduction of mutant C1ql2 in Bcl11b cKOs, KD of C1ql2 in WT mice, and Nrxn123 cKO) and reached similar conclusions. Together, this strongly supports that the observed phenotypes occur as a result of the physiological function of the proteins involved in the described mechanism and not due to interference of the AAVs with these biological processes. We have now addressed this point in the main text module of the revised ms.

The authors claim that the viral re-introduction "restored C1ql2 protein expression to control levels.This is misleading given that the mean of the data is 2.5x the control (Figure 1d and also see Figure 6c). The low n and large variance are a problem for these data. Moreover, they are marked ns but the authors should report p values for these. At the least, this likely large overexpression and variability should be acknowledged. In addition, the use of clipped bands on Western blots should be avoided. Please show the complete protein gel in primary figures of supplemental information.

We agree with reviewer #2 that C1ql2 expression after its re-introduction in Bcl11b cKO mice was higher compared to controls and that this should be taken into consideration for proper interpretation of the data. To address this, based also on the suggestion of reviewer #3 point 1 below, we overexpressed C1ql2 in DG neurons of control animals. We found no changes in synaptic vesicle organization upon C1ql2 over-expression compared to controls. This further supports that the observed effect upon rescue of C1ql2 expression in Bcl11b cKOs is due to the physiological function of C1ql2 and not as result of the overexpression. These data are included in supplementary figure 2g-j and are described in detail in the results part of the revised manuscript.

Additionally, we looked at the effects of C1ql2 overexpression in Bcl11b cKO DGN on basal synaptic transmission. We plotted fEPSP slopes versus fiber volley amplitudes, measured in slices from rescue animals, as we had previously done for the control and Bcl11b cKO (Author response image 1a). Although regression analysis revealed a trend towards steeper slopes in the rescue mice (Author response image 1a and b), the observation did not prove to be statistically significant, indicating that C1ql2 overexpression in Bcl11b cKO animals does not strongly alter basal synaptic transmission at MFS. Overall, our previous and new findings support that the observed effects of the C1ql2 rescue are not caused by the artificially elevated levels of C1ql2, as compared to controls, but are rather a result of the physiological function of C1ql2.

Following the suggestion of reviewer #2 all western blot clipped bands were exchanged for images of the full blot. This includes figures 1c, 4c, 6b and supplementary figure 2g of the revised manuscript. P-value for Figure 1d has now been included.

**Author response image 1. sa4fig1:** C1ql2 reintroduction in Bcl11b cKO DGN does not significantly alter basal synaptic transmission at mossy fiber-CA3 synapses. a Input-output curves generated by plotting fEPSP slope against fiber volley amplitude at increasing stimulation intensities. b Quantification of regression line slopes for input-output curves for all three conditions. Control+EGFP, 35 slices from 16 mice; Bcl11b cKO+EGFP, 32 slices from 14 mice; Bcl11b cKO+EGFP-2A-C1ql2, 22 slices from 11 mice. The data are presented as means, error bars represent SEM. Kruskal-Wallis test (non-parametric ANOVA) followed by Dunn’s post hoc pairwise comparisons. p=0.106; ns, not significant.

Measurement of EM micrographs: As prior work suggested that MF synapse structure is disrupted the authors should report active zone length as this may itself affect "synapse score" defined by the number of vesicles docked. More concerning is that the example KO micrographs seem to have lost all the densely clustered synaptic vesicles that are away from the AZ in normal MF synapses e.g. compare control and KO terminals in Fig 2a or 6f or 7f. These terminals look aberrant and suggest that the important measure is not what is docked but what is present in the terminal cytoplasm that normally makes up the reserve pool. This needs to be addressed with further analysis and modifications to the manuscript.

As requested by reviewer #2 we analyzed and reported in the revised manuscript the active zone length. We found that the active zone length remained unchanged in all conditions (control/Bcl11b cKO/C1ql2 rescue, WT/C1ql2 KD, control/K262E and control/Nrxn123 cKO), strengthening our results that the described Bcl11b/C1ql2/Nrxn3 mechanism is involved in the recruitment of synaptic vesicles. These data have been included in supplementary figures 2c, 4h, 5f and 6g and are described in the results part of the revised manuscript.

We want to clarify that the synapse score is not defined by the number of docked vesicles to the plasma membrane. The synapse score, which is described in great detail in our materials and methods part and has been previously published (De Bruyckere et al., 2018), rates MFS based on the number of synaptic vesicles and their distance from the active zone and was designed according to previously described properties of the vesicle pools at the MFS. The EM micrographs refer to the general misdistribution of SV in the proximity of MFS. Upon revision of the manuscript, we made sure that this was clearly stated in the main text module to avoid further confusion.

The study also presents correlated changes in MF LTP in Bcl11b KO which are rescued by C1ql2 expression. It is not clear whether the structural and functional deficits are causally linked and this should be made clearer in the manuscript. It is also not apparent why this functional measure was chosen as it is unlikely that C1ql2 plays a direct role in presynaptic plasticity mechanisms that are through a cAMP/ PKA pathway and likely disrupted LTP is due to dysfunctional synapses rather than a specific LTP effect.

The inclusion of functional experiments in this and our previous study (de Bruyckere et al., 2018) was first and foremost intended to determine whether the structural alterations observed at MFB disrupt MFS signaling. From the signaling properties we tested, basal synaptic transmission (this study) and short-term potentiation (de Bruyckere et al., 2018) were unaltered by Bcl11b KO, whereas MF LTP was found to be abolished (de Bruyckere et al., 2018). Indeed, because MF LTP largely depends on presynaptic mechanisms, including the redistribution of the readily releasable pool and recruitment of new active zones (Orlando et al., 2021; Vandael et al., 2020), it appears to be particularly sensitive to the specific structural changes we observed. We therefore believe that it is valuable information that MF LTP is affected in Bcl11b cKO animals - it conveys a direct proof for the functional importance of the observed morphological alterations, while basic transmission remains largely normal. Furthermore, it subsequently provided a functional marker for testing whether the reintroduction of C1ql2 in Bcl11b cKO animals or the KD of C1ql2 in WT animals can functionally recapitulate the control or the Bcl11b KO phenotype, respectively.

We fully agree with the reviewer that C1ql2 is unlikely to directly participate in the cAMP/PKA pathway and that the ablation of C1ql2 likely disrupts MF LTP through an alternative mode of action. Our original wording in the paragraph describing the results of the forskolin-induced LTP experiment might have overstressed the importance of the cAMP pathway. We have now rephrased that paragraph to better describe the main idea behind the forskolin experiment, namely to circumvent the initial Ca2+ influx in order to test whether deficient presynaptic Ca2+ channel/KAR signaling might be responsible for the loss of LTP in Bcl11b cKO. The results are strongly indicative of a downstream mechanism and further investigation is needed to determine the specific mechanisms by which C1ql2 regulates MFLTP, especially in light of the result that C1ql2.K262E rescued LTP, while it was unable to rescue the SV recruitment at the MF presynapse. This raises the possibility that C1ql2 can influence MF-LTP through additional, yet uncharacterized mechanisms, independent of SV recruitment. As such, a causal link between the structural and functional deficits remains tentative and we have now emphasized that point by adding a respective sentence to the discussion of our revised manuscript. Nevertheless, we again want to stress that the main rationale behind the LTP experiments was to assess the functional significance of structural changes at MFS and not to elucidate the mechanisms by which MF LTP is established.

The authors should consider measures that might support the role of Bcl11b targets in SV recruitment during the depletion of synapses or measurements of the readily releasable pool size that would complement their findings in structural studies.

We fully agree that functional measurements of the readily releasable pool (RRP) size would be a valuable addition to the reported redistribution of SV in structural studies. We have, in fact, attempted to use high-frequency stimulus trains in both field and single-cell recordings (details on single-cell experiments are described in the response to point 8) to evaluate potential differences in RRP size between the control and Bcl11b KO (Figure for reviewers 2a and b). Under both recording conditions we see a trend towards lower values of the intersection between a regression line of late responses and the y-axis. This could be taken as an indication of slightly smaller RRP size in Bcl11b mutant animals compared to controls. However, due to several technical reasons we are extremely cautious about drawing such far-reaching conclusions based on these data. At most, they suffice to conclude that the availability of release-ready vesicles in the KO is likely not dramatically smaller than in the control.

The primary issue with using high-frequency stimulus trains for RRP measurements at MFS is the particularly low initial release probability (Pr) at these synapses. This means that a large number of stimulations is required to deplete the RRP. As the RRP is constantly replenished, it remains unclear when steady state responses are reached (reviewed by Kaeser and Regehr, 2017). This is clearly visible in our single-cell recordings (Author response image 2b), which were additionally complicated by prominent asynchronous release at later stages of the stimulus train and by a large variability in the shapes of cumulative amplitude curves between cells. In contrast, while the cumulative amplitude curves for field potential recordings do reach a steady state (Author response image 2a), field potential recordings in this context are not a reliable substitute for single cell or, in the case of MFB, singlebouton recordings. Postsynaptic cells in field potential recordings are not clamped, meaning that the massive release of glutamate due to continuous stimulation depolarizes the postsynaptic cells and reduces the driving force for Na+, irrespective of depletion of the RRP. This is supported by the fact that we consistently observed a recovery of fEPSP amplitudes later in the trains where RRP had presumably been maximally depleted. In summary, high-frequency stimulus trains at the field potential level are not a valid and established technique for estimating RRP size at MFS.

Specialized laboratories have used highly advanced techniques, such as paired recordings between individual MFB and postsynaptic CA3 pyramidal cells, to estimate the RRP size of MFB (Vandael et al., 2020). These approaches are outside the scope of our present study which, while elucidating functional changes following Bcl11b depletion and C1ql2 rescue, does not aim to provide a high-end biophysical analysis of the presynaptic mechanisms involved.

**Author response image 2. sa4fig2:** Estimation of RRP size using high-frequency stimulus trains at mossy fiber-CA3 synapses. a Results from field potential recordings. Cumulative fEPSP amplitude in response to a train of 40 stimuli at 100 Hz. All subsequent peak amplitudes were normalized to the amplitude of the first peak. Data points corresponding to putative steady state responses were fit with linear regression (RRP size is indirectly reflected by the intersection of the regression line with the yaxis). Control+EGFP, 6 slices from 5 mice; Bcl11b cKO+EGFP, 6 slices from 3 mice. b Results from single-cell recordings. Cumulative EPSC amplitude in response to a train of 15 stimuli at 50 Hz. The last four stimuli were fit with linear regression. Control, 5 cells from 4 mice; Bcl11b cKO, 3 cells from 3 mice. Note the shallow onset of response amplitudes and the subsequent frequency potentiation. Due to the resulting increase in slope at higher stimulus numbers, intersection with the y-axis occurs at negative values. The differences shown were not found to be statistically significant; unpaired t-test or Mann-Whitney U-test.

Bcl11b KO reduces the number of synapses, yet the I-O curve reported in Supp Fig 2 is not changed. How is that possible? This should be explained.

We agree with reviewer #2– this apparent discrepancy has indeed struck us as a counterintuitive result. It might be that synapses that are preferentially eliminated in Bcl11b cKO are predominantly silent or have weak coupling strength, such that their loss has only a minimal effect on basal synaptic transmission. Although perplexing, the result is fully supported by our single-cell data which shows no significant differences in MF EPSC amplitudes recorded from CA3 pyramidal cells between controls and Bcl11b mutants (Author response image 3; please see the response below for details and also our response to Reviewer #1 question 2).

Matsuda et al DOI: 10.1016/j.neuron.2016.04.001 previously reported that C1ql2 organizes MF synapses by aligning postsynaptic kainate receptors with presynaptic elements. As this may have consequences for the functional properties of MF synapses including their plasticity, the authors should report whether they see deficient postsynaptic glutamate receptor signaling in the Bcl11b KO and rescue in the C1ql2 re-expression.

We agree that the study by Matsuda et al. is of key importance for our present work. Although MF LTP is governed by presynaptic mechanisms and we previously did not see differences in short-term plasticity between the control and Bcl11b cKO (De Bruyckere et al., 2018), the clustering of postsynaptic kainate receptors by C1ql2 is indeed an important detail that could potentially alter synaptic signaling at MFS in Bcl11b KO. We, therefore, re-analyzed previously recorded single-cell data by performing a kinetic analysis on MF EPSCs recorded from CA3 pyramidal cells in control and Bcl11b cKO mice (Figure for reviewers 3a) to evaluate postsynaptic AMPA and kainate receptor responses in both conditions. We took advantage of the fact that AMPA receptors deactivate roughly 10 times faster than kainate receptors, allowing the contributions of the two receptors to mossy fiber EPSCs to be separated (Castillo et al., 1997 and reviewed by Lerma, 2003). We fit the decay phase of the second (larger) EPSC evoked by paired-pulse stimulation with a double exponential function, yielding a fast and a slow component, which roughly correspond to the fractional currents evoked by AMPA and kainate receptors, respectively. Analysis of both fast and slow time constants and the corresponding fractional amplitudes revealed no significant differences between controls and Bcl11b mutants (Figure for reviewers 3e-h), indicating that both AMPA and kainate receptor signaling is unaffected by the ablation of C1ql2 following Bcl11b KO.

Importantly, MF EPSC amplitudes evoked by the first and the second pulse (Author response image 3b), paired-pulse facilitation (Author response image 3c) and failure rates (Author response image 3d) were all comparable between controls and Bcl11b mutants. These results further corroborate our observations from field recordings that basal synaptic transmission at MFS is unaltered by Bcl11b KO.

We note that the results from single cell recordings regarding basal synaptic transmission merely confirm the observations from field potential recordings, and that the attempted measurement of RRP size at the single cell level was not successful. Thus, our single-cell data do not add new information about the mechanisms underlying the effects of Bcl11b-deficiency and we therefore decided not to report these data in the manuscript.

**Author response image 3. sa4fig3:** Basal synaptic transmission at mossy fiber-CA3 synapses is unaltered in Bcl11b cKO mice. a Representative average trace (20 sweeps) recorded from CA3 pyramidal cells in control and Bcl11b cKO mice at minimal stimulation conditions, showing EPSCs in response to paired-pulse stimulation (PPS) at an interstimulus interval of 40 ms. The signal is almost entirely blocked by the application of 2 μM DCG-IV (red). b Quantification of MF EPSC amplitudes in response to PPS for both the first and the second pulse. c Ratio between the amplitude of the second over the first EPSC. d Percentage of stimulation events resulting in no detectable EPSCs for the first pulse. Events <5 pA were considered as noise. e Fast decay time constant obtained by fitting the average second EPSC with the following double exponential function: I(t)=Afaste−t/τfast+Aslowe−t/τslow+C, where I is the recorded current amplitude after time t, Afast and Aslow represent fractional current amplitudes decaying with the fast (τfast) and slow (τslow) time constant, respectively, and C is the offset. Starting from the peak of the EPSC, the first 200 ms of the decaying trace were used for fitting. f Fractional current amplitude decaying with the fast time constant. g-h Slow decay time constant and fractional current amplitude decaying with the slow time constant. For all figures: Control, 8 cells from 4 mice; Bcl11b cKO, 8 cells from 6 mice. All data are presented as means, error bars indicate SEM. None of the differences shown were found to be statistically significant; Mann-Whitney U-test for nonnormally and unpaired t-test for normally distributed data.

**Reviewer #3 (Public Review):**
Overall, this is a strong manuscript that uses multiple current techniques to provide specific mechanistic insight into prior discoveries of the contributions of the Bcl11b transcription factor to mossy fiber synapses of dentate gyrus granule cells. The authors employ an adult deletion of Bcl11b via Tamoxifen-inducible Cre and use immunohistochemical, electron microscopy, and electrophysiological studies of synaptic plasticity, together with viral rescue of C1ql2, a direct transcriptional target of Bcl11b or Nrxn3, to construct a molecular cascade downstream of Bcl11b for DG mossy fiber synapse development. They find that C1ql2 re-expression in Bcl11b cKOs can rescue the synaptic vesicle docking phenotype and the impairments in MF-LTP of these mutants. They also show that C1ql2 knockdown in DG neurons can phenocopy the vesicle docking and plasticity phenotypes of the Bcl11b cKO. They also use artificial synapse formation assays to suggest that C1ql2 functions together with a specific Nrxn3 splice isoform in mediating MF axon development, extending these data with a C1ql2-K262E mutant that purports to specifically disrupt interactions with Nrxn3. All of the molecules involved in this cascade are disease-associated and this study provides an excellent blueprint for uncovering downstream mediators of transcription factor disruption. Together this makes this work of great interest to the field. Strengths are the sophisticated use of viral replacement and multi-level phenotypic analysis while weaknesses include the linkage of C1ql2 with a specific Nrxn3 splice variant in mediating these effects.Here is an appraisal of the main claims and conclusions:1. C1ql2 is a downstream target of Bcl11b which mediates the synaptic vesicle recruitment and synaptic plasticity phenotypes seen in these cKOs. This is supported by the clear rescue phenotypes of synapse anatomy (Fig.2) and MF synaptic plasticity (Fig.3). One weakness here is the absence of a control assessing over-expression phenotypes of C1ql2. It's clear from Fig.1D that viral rescue is often greater than WT expression (totally expected). In the case where you are trying to suppress a LoF phenotype, it is important to make sure that enhanced expression of C1ql2 in a WT background does not cause your rescue phenotype. A strong overexpression phenotype in WT would weaken the claim that C1ql2 is the main mediator of the Bcl11b phenotype for MF synapse phenotypes.

As suggested by reviewer #3, we carried out C1ql2 over-expression experiments in control animals. We show that the over-expression of C1ql2 in the DG of control animals had no effect on the synaptic vesicle organization in the proximity of MFS. This further supports that the observed effect upon rescue of C1ql2 expression in Bcl11b cKOs is due to the physiological function of C1ql2 and not a result of the artificial overexpression. These data are now included in supplementary figure 2g-j and are described in detail in the results part of the revised manuscript. Please also see response to point 3 of reviewer #2.

1. Knockdown of C1ql2 via 4 shRNAs is sufficient to produce the synaptic vesicle recruitment and MFLTP phenotypes. This is supported by clear effects in the shRNA-C1ql2 groups as compared to nonsense-EGFP controls. One concern (particularly given the use of 4 distinct shRNAs) is the potential for off-target effects, which is best controlled for by a rescue experiment with RNA insensitive C1ql2 cDNA as opposed to nonsense sequences, which may not elicit the same off-target effects.

We agree with reviewer #3 that the usage of shRNAs could potentially create unexpected off-target effects and that the introduction of a shRNA-insensitive C1ql2 in parallel to the expression on the shRNA cassette would be a very effective control experiment. However, the suggested experiment would require an additional 6 months (2 months for AAV production, 2-3 months from animal injection to sacrifice and 1-2 months for EM imaging/analysis and LTP measurements) and a high number of additional animals (minimum 8 for EM and 8 for LTP measurements). We note here, that before the production of the shRNA-C1ql2 and the shRNA-NS, the individual sequences were systematically checked for off-target bindings on the murine exome with up to two mismatches and presented with no other target except the proposed (C1ql2 for shRNA-C1ql2 and no target for shRNA-NS). Taking into consideration our in-silico analysis, we feel that the interpretation of our findings is valid without this (very reasonable) additional control experiment.

1. C1ql2 interacts with Nrxn3(25b+) to facilitate MF terminal SV clustering. This claim is theoretically supported by the HEK cell artificial synapse formation assay (Fig.5), the inability of the K262-C1ql2 mutation to rescue the Bcl11b phenotype (Fig.6), and the altered localization of C1ql2 in the Nrxn1-3 deletion mice (Fig.7). Each of these lines of experimental evidence has caveats that should be acknowledged and addressed. Given the hypothesis that C1ql2 and Nrxn3b(25b) are expressed in DG neurons and work together, the heterologous co-culture experiment seems strange. Up till now, the authors are looking at pre-synaptic function of C1ql2 since they are re-expressing it in DGNs. The phenotypes they are seeing are also pre-synaptic and/or consistent with pre-synaptic dysfunction. In Fig.5, they are testing whether C1ql2 can induce pre-synaptic differentiation in trans, i.e. theoretically being released from the 293 cells "post-synaptically". But the post-synaptic ligands (Nlgn1 and GluKs) are not present in the 293 cells, so a heterologous synapse assay doesn't really make sense here. The effect that the authors are seeing likely reflects the fact that C1ql2 and Nrxn3 do bind to each other, so C1ql2 is acting as an artificial post-synaptic ligand, in that it can cluster Nrxn3 which in turn clusters synaptic vesicles. But this does not test the model that the authors propose (i.e. C1ql2 and Nrxn3 are both expressed in MF terminals). Perhaps a heterologous assay where GluK2 is put into HEK cells and the C1ql2 and Nrxn3 are simultaneously or individually manipulated in DG neurons?

C1ql2 is expressed by DG neurons and is then secreted in the MFS synaptic cleft, while Nrxn3, that is also expressed by DG neurons, is anchored at the presynaptic side. In our work we used the well established co-culture system assay and cultured HEK293 cells secreting C1ql2 (an IgK secretion sequence was inserted at the N-terminus of C1ql2) together with hippocampal neurons expressing Nrxn3(25b+). We used the HEK293 cells as a delivery system of secreted C1ql2 to the neurons to create regions of high concentration of C1ql2. By interfering with the C1ql2-Nrxn3 interaction in this system either by expression of the non-binding mutant C1ql2 variant in the HEK cells or by manipulating Nrxn expression in the neurons, we could show that C1ql2 binding to Nrxn3(25b+) is necessary for the accumulation of vGlut1. However, we did not examine and do not claim within our manuscript that the interaction between C1ql2 and Nrxn3(25b+) induces presynaptic differentiation. Our experiment only aimed to analyze the ability of C1ql2 to cluster SV through interaction with Nrxn3. Moreover, by not expressing potential postsynaptic interaction partners of C1ql2 in our system, we could show that C1ql2 controls SV recruitment through a purely presynaptic mechanism. Co-culturing GluK2-expressing HEK cells with simultaneous manipulation of C1ql2 and/or Nrxn3 in neurons would not allow us to appropriately answer our scientific question, but rather focus on the potential synaptogenic function of the Nrxn3/C1ql2/GluK2 complex and the role of the postsynaptic ligand in it. Thus, we feel that the proposed experiment, while very interesting in characterization of additional putative functions of C1ql2, may not provide additional information for the point we were addressing. In the revised manuscript we tried to make the aim and methodological approach of this set of experiments more clear.

1. K262-C1ql2 mutation blocks the normal rescue through a Nrxn3(25b) mechanism (Fig.6). The strength of this experiment rests upon the specificity of this mutation for disrupting Nrxn3b binding (presynaptic) as opposed to any of the known postsynaptic C1ql2 ligands such as GluK2. While this is not relevant for interpreting the heterologous assay (Fig.5), it is relevant for the in vivo phenotypes in Fig.6. Similar approaches as employed in this paper can test whether binding to other known postsynaptic targets is altered by this point mutation.

It has been previously shown that C1ql2 together with C1ql3 recruit postsynaptic GluK2 at the MFS. However, loss of just C1ql2 did not affect the recruitment of GluK2, which was disrupted only upon loss of both C1ql2 and C1ql3 (Matsuda et al., 2018). In our study we demonstrate a purely presynaptic function of C1ql2 through Nrxn3 in the synaptic vesicle recruitment. This function is independent of C1ql3, as C1ql3 expression is unchanged in all of our models and its over-expression did not compensate for C1ql2 functions (Fig. 2, 3a-c). Our in vitro experiments also reveal that C1ql2 can recruit both Nrxn3 and vGlut1 in the absence of any known postsynaptic C1ql2 partner (KARs and BAI3; Fig.5; please also see response above). Furthermore, we have now performed a kinetic analysis on single-cell data which we had previously collected to evaluate postsynaptic AMPA and kainate receptor responses in both the control and Bcl11b KO. Our analysis reveals no significant differences in postsynaptic current kinetics, making it unlikely that AMPA and kainate receptor signaling is altered upon the loss of C1ql2 following Bcl11b cKO (Author response image 3e-h; please also see our response to reviewer #2 point 8). Thus, we have no experimental evidence supporting the idea that a loss of interaction between C1ql2.K262E and GluK2 would interfere with the examined phenotype. However, to exclude that the K262E mutation disrupts interaction between C1ql2 and GluK2, we performed co-immunoprecipitation from protein lysate of HEK293 cells expressing GluK2myc-flag and GFP-C1ql2 or GluK2-myc-flag and GFP-K262E and could show that both C1ql2 and K262E had GluK2 bound when precipitated. These data are included in supplementary figure 5k of the revised manuscript.

1. Altered localization of C1ql2 in Nrxn1-3 cKOs. These data are presented to suggest that Nrx3(25b) is important for localizing C1ql2 to the SL of CA3. Weaknesses of this data include both the lack of Nrxn specificity in the triple a/b KOs as well as the profound effects of Nrxn LoF on the total levels of C1ql2 protein. Some measure that isn't biased by this large difference in C1ql2 levels should be attempted (something like in Fig.1F).

We acknowledge that the lack of specificity in the Nrxn123 model makes it difficult to interpret our data. We have now examined the mRNA levels of Nrxn1 and Nrxn2 upon stereotaxic injection of Cre in the DG of Nrxn123flox/flox animals and found that Nrxn1 was only mildly reduced. At the same time Nrxn2 showed a tendency for reduction that was not significant (data included in supplementary figure 6a of revised manuscript). Only Nrxn3 expression was strongly suppressed. Of course, this does not exclude that the mild reduction of Nrxn1 and Nrxn2 interferes with the C1ql2 localization at the MFS. We further examined the mRNA levels of C1ql2 in control and Nrxn123 mutants to ensure that the observed changes in C1ql2 protein levels at the MFS are not due to reduced mRNA expression and found no changes (data are included in supplementary figure 6b of the revised manuscript), suggesting that overall protein C1ql2 expression is normal.

The reduced C1ql2 fluorescence intensity at the MFS was first observed when non-binding C1ql2 variant K262E was introduced to Bcl11b cKO mice that lack endogenous C1ql2 (Fig.6). In these experiments, we found that despite the overall high protein levels of C1ql2.K262E in the hippocampus (Fig. 6c), its fluorescence intensity at the SL was significantly reduced compared to WT C1ql2 (Fig. 6d-e). The remaining signal of the C1ql2.K262E at the SL was equally distributed and in a punctate form, similar to WT C1ql2. Together, this suggests that loss of C1ql2-Nrxn3 interaction interferes with the localization of C1ql2 at the MFS, but not with the expression of C1ql2. Of course, this does not exclude that other mechanisms are involved in the synaptic localization of C1ql2, beyond the interaction with Nrxn3, as both the mutant C1ql2 in Bcl11b cKO and the endogenous C1ql2 in Nrxn123 cKOs show residual immunofluorescence at the SL. Further studies are required to determine how C1ql2-Nrxn3 interaction regulates C1ql2 localization at the MFS.

**Reviewer #1 (Recommendations For The Authors):**
In addition to addressing the comments below, this study would benefit significantly from providing insight and discussion into the relevant potential postsynaptic signaling components controlled exclusively by C1ql2 (postsynaptic kainate receptors and the BAI family of proteins).

We have now performed a kinetic analysis on single-cell data that we had previously collected to evaluate postsynaptic AMPA and kainate receptor responses in both the control and Bcl11b cKO. Our analysis reveals no significant differences in postsynaptic current kinetics, making it unlikely that AMPA and kainate receptor signaling differ between controls and upon the loss of C1ql2 following Bcl11b cKO (Author response image 3e-h; please also see our response to Reviewer #2 point 8). This agrees with previous findings that C1ql2 regulates postsynaptic GluK2 recruitment together with C1ql3 and only loss of both C1ql2 and C1ql3 results in a disruption of KAR signaling (Matsuda et al., 2018). In our study we demonstrate a purely presynaptic function of C1ql2 through Nrxn3 in the synaptic vesicle recruitment. This function is independent of C1ql3, as C1ql3 expression is unchanged in all of our models and its over-expression did not compensate for C1ql2 functions (Fig. 2, 3a-c). Our in vitro experiments also reveal that C1ql2 can recruit both Nrxn3 and vGlut1 in the absence of any known postsynaptic C1ql2 partner (KARs and BAI3; Fig.5; please also see our response to reviewer #3 point 4 above). We believe that further studies are needed to fully understand both the pre- and the postsynaptic functions of C1ql2. Because the focus of this manuscript was on the role of the C1ql2-Nrxn3 interaction and our investigation on postsynaptic functions of C1ql2 was incomplete, we did not include our findings on postsynaptic current kinetics in our revised manuscript. However, we increased the discussion on the known postsynaptic partners of C1ql2 in the revised manuscript to increase the interpretability of our results.

Major Comments:The authors demonstrate that the ultrastructural properties of presynaptic boutons are altered after Bcl11b KO and C1ql2 KD. However, whether C1ql2 functions as part of a tripartite complex and the identity of the postsynaptic receptor (BAI, KAR) should be examined.

Matsuda and colleagues have nicely demonstrated in their 2016 (Neuron) study that C1ql2 is part of a tripartite complex with presynaptic Nrxn3 and postsynaptic KARs. Moreover, they demonstrated that C1ql2, together with C1ql3, recruit postsynaptic KARs at the MFS, while the KO of just C1ql2 did not affect the KAR localization. In our study we demonstrate a purely presynaptic function of C1ql2 through Nrxn3 in the synaptic vesicle recruitment. This function is independent of C1ql3, as C1ql3 expression is unchanged in all of our models and its over-expression did not compensate for C1ql2 functions (Fig. 2, 3a-c). Our in vitro experiments also reveal that C1ql2 is able to recruit both Nrxn3 and vGlut1 in the absence of any known postsynaptic C1ql2 partner (Fig. 5; please also see our response to reviewer #3 point 4 above). Moreover, we were able to show that the SV recruitment depends on C1ql2 interaction with Nrxn3 through the expression of a non-binding C1ql2 (Fig. 6) that retains the ability to interact with GluK2 (supplementary figure 5k of revised manuscript) or by KO of Nrxns (Fig. 7). Furthermore, we have now performed a kinetic analysis on single-cell data which we had previously collected to evaluate postsynaptic AMPA and kainate receptor responses in both the control and Bcl11b cKO. Our analysis reveals no significant differences in postsynaptic current kinetics, making it unlikely that AMPA and kainate receptor signaling differ between controls and Bcl11b mutants (Author response image 3e-h; please also see our response to Reviewer #2 question 8). Together, we have no experimental evidence so far that would support that the postsynaptic partners of C1ql2 are involved in the observed phenotype. While it would be very interesting to characterize the postsynaptic partners of C1ql2 in depth, we feel this would be beyond the scope of the present study.

Figure 1f: For a more comprehensive understanding of the Bcl11b KO phenotype and the potential role for C1ql2 on MF synapse number, a complete quantification of vGlut1 and Homer1 for all conditions (Supplement Figure 2e) should be included in the main text.

In our study we focused on the role of C1ql2 in the structural and functional integrity of the MFS downstream of Bcl11b. Bcl11b ablation leads to several phenotypes in the MFS that have been thoroughly described in our previous study (De Bruyckere et al., 2018). As expected, re-expression of C1ql2 only partially rescued these phenotypes, with full recovery of the SV recruitment (Fig. 2) and of the LTP (Fig. 3), but had no effect on the reduced numbers of MFS nor the structural complexity of the MFB created by the Bcl11b KO (supplementary figure 2d-f of revised manuscript). We understand that including the quantification of vGlut1 and Homer1 co-localization in the main figures would help with a better understanding of the Bcl11b mutant phenotype. However, in our manuscript we investigate C1ql2 as an effector of Bcl11b and thus we focus on its functions in SV recruitment and LTP. As we did not find a link between C1ql2 and the number of MFS/MFB upon re-expression of C1ql2 in Bcl11b cKO or now also in C1ql2 KD (see response to comment #4 below), we believe it is more suitable to present these data in the supplement.

Figure 3/4: Given the striking reduction in the numbers of synapses (Supplement Figure 2e) and docked vesicles (Figure 2d) in the Bcl11b KO and C1ql2 KD (Figure 4e-f), it is extremely surprising that basal synaptic transmission is unaffected (Supplement Figure 2g). The authors should determine the EPSP input-output relationship following C1ql2 KD and measure EPSPs following trains of stimuli at various high frequencies.

We fully acknowledge that this is an unexpected result. It is, however, well feasible that the modest displacement of SV fails to noticeably influence basal synaptic transmission. This would be the case, for example, if only a low number of vesicles are released by single stimuli, in line with the very low initial Pr at MFS. In contrast, the reduction in synapse numbers in the Bcl11b mutant might indeed be expected to reflect in the input-output relationship. It is possible, however, that synapses that are preferentially eliminated in Bcl11b cKO are predominantly silent or have weak coupling strength, such that their loss has only a minimal effect on basal synaptic transmission. Finally, we cannot exclude compensatory mechanisms (homeostatic plasticity) at the remaining synapses. A detailed analysis of these potential mechanisms would be a whole project in its own right.

As additional information, we can say that the largely unchanged input-output-relation in Bcl11b cKO is also present in the single-cell level data (Author response image 3; details on single-cell experiments are described in the response to Reviewer #2 point 8).

As suggested by the reviewer, we have now additionally analyzed the input-output relationship following C1ql2 KD and again did not observe any significant difference between control and KD animals. We have incorporated the respective input-output curves into the revised manuscript under Supplementary figure 3c-d.

Figure 4: Does C1ql2 shRNA also reduce the number of MFBs? This should be tested to further identify C1ql2-dependent and independent functions.

As requested by reviewer #1 we quantified the number of MFBs upon C1ql2 KD. We show that C1ql2 KD in WT animals does not alter the number of MFBs. The data are presented in supplementary figure 4d of the revised manuscript. Re-expression of C1ql2 in Bcl11b cKO did not rescue the loss of MFS created by the Bcl11b mutation. Moreover, C1ql2 re-expression did not rescue the complexity of the MFB ultrastructure perturbed by the Bcl11b ablation. Together, this suggests that Bcl11b regulates MFs maintenance through additional C1ql2-independent pathways. In our previously published work (De Bruyckere et al., 2018) we identified and discussed in detail several candidate genes such as Sema5b, Ptgs2, Pdyn and Penk as putative effectors of Bcl11b in the structural and functional integrity of MFS (please also see response to reviewer #2- point 1 of public reviews).

Figure 5: Clarification is required regarding the experimental design of the HEK/Neuron co-culture: 1. C1ql2 is a secreted soluble protein - how is the protein anchored to the HEK cell membrane to recruit Nrxn3(25b+) binding and, subsequently, vGlut1?

C1ql2 was secreted by the HEK293 cells through an IgK signaling peptide at the N-terminus of C1ql2. The high concentration of C1ql2 close to the secretion site together with the sparse coculturing of the HEK293 cells on the neurons allows for the quantification of accumulation of neuronal proteins. We have now described the experimental conditions in greater detail in the main text module of the revised manuscript

1. Why are the neurons transfected and not infected? Transfection efficiency of neurons with lipofectamine is usually poor (1-5%; Karra et al., 2010), while infection of neurons with lentiviruses or AAVs encoding cDNAs routinely are >90% efficient. Thus, interpretation of the recruitment assays may be influenced by the density of neurons transfected near a HEK cell.

We agree with reviewer #1 that viral infection of the neurons would have been a more effective way of expressing our constructs. However, due to safety allowances in the used facility and time limitation at the time of conception of this set of experiments, a lipofectamine transfection was chosen.

However, as all of our examined groups were handled in the same way and multiple cells from three independent experiments were examined for each experimental set, we believe that possible biases introduced by the transfection efficiency have been eliminated and thus have trust in our interpretation of these results.

1. Surface labeling of HEK cells for wild-type C1ql2 and K262 C1ql2 would be helpful to assess the trafficking of the mutant.

We recognize that potential changes to the trafficking of C1ql2 caused by the K262E mutation would be important to characterize, in light of the reduced localization of the mutant protein at the SL in the in vivo experiments (Fig. 6e). In our culture system, C1ql2 and K262E were secreted by the HEK cells through insertion of an IgK signaling peptide at the N-terminus of the myc-tagged C1ql2/K262E. Thus, trafficking analysis on this system would not be informative, as the system is highly artificial compared to the in vivo model. Further studies are needed to characterize C1ql2 trafficking in neurons to understand how C1ql2-Nrxn3 interaction regulates the localization of C1ql2. However, labeling of the myc-tag in C1ql2 or K262E expressing HEK cells of the co-culture model reveals a similar signal for the two proteins (Fig. 5a,c). Nrxn-null mutation in neurons co-cultured with C1ql2-expressing HEK cells disrupted C1ql2 mediated vGlut1 accumulation in the neurons. Selective expression of Nrxn3(25b) in the Nrxn-null neurons restored vGlut1 clustering was (Fig. 5e-f). Together, these data suggest that it is the interaction between C1ql2 and Nrxn3 that drives the accumulation of vGlut1.

Figure 6: Bcl11b KO should also be included in 6f-h.

As suggested by reviewer #1, we included the Bcl11b cKO in figures 6f-h and in corresponding supplementary figures 5c-j.

Figure 7b: What is the abundance of mRNA for Nrxn1 and Nrxn2 as well as the abundance of Nrxns after EGFP-Cre injection into DG?

We addressed this point raised by reviewer #1 by quantifying the relative mRNA levels of Nrxn1 and Nrxn2 via qPCR upon Nrxn123 mutation induction with EGFP-Cre injection. We have now examined the mRNA levels of Nrxn1 and Nrxn2 upon stereotaxic injection of Cre in the DG of Nrxn123flox/flox animals and found that Nrxn1 was only mildly reduced. At the same time Nrxn2 showed a tendency for reduction that was not significant. The data are presented in supplementary figure 6a of the revised maunscript.

Minor Comments for readability:Synapse score is referred to frequently in the text and should be defined within the text for clarification.'n' numbers should be better defined in the figure legends. For example, for protein expression analysis in 1c, n=3. Is this a biological or technical triplicate? For electrophysiology (e.g. 3c), does"n=7" reflect the number of animals or the number of slices? n/N (slices/animals) should be presented.Figure 7a: Should the diagrams of the cre viruses be EGFP-Inactive or active Cre and not CRE-EGFP as shown in the diagram?Figure 7b: the region used for the inset should be identified in the larger image.

All minor points have been fixed in the revised manuscript according to the suggestions.

**Reviewer #3 (Recommendations For The Authors):**
-Please describe the 'synapse score' somewhere in the text - it is too prominently featured to not have a clear description of what it is.

The description of the synapse score has been included in the main text module of the revised manuscript.

-The claim that Bcl11b controls SV recruitment "specifically" through C1ql2 is a bit stronger than is warranted by the data. Particularly given that C1ql2 is expressed at 2.5X control levels in their rescue experiments. See pt.2

Please see response to reviewer #3 point 1 of public reviews. To address this, we over-expressed C1ql2 in control animals and found no changes in the synaptic vesicle distribution (supplementary figure 2g-j of revised manuscript). This supports that the observed rescue of synaptic vesicle recruitment by re-expression of C1ql2 is due to its physiological function and not due to the artificially elevated protein levels. Of course, we cannot exclude the possibility that other, C1ql2-independent, mechanisms also contribute to the SV recruitment downstream of Bcl11b. Our data from the C1ql2 rescue, C1ql2 KD, the in vitro experiments and the interruption of C1ql2-Nrxn3 in vivo, strongly suggest C1ql2 to be an important regulator of SV recruitment.

-Does Bcl11b regulate Nrxn3 expression? Considering the apparent loss of C1ql2 expression in the Nrxn KO mice, this is an important detail.

We agree with reviewer #3 that this is an important point. We have previously done differential transcriptomics from DG neurons of Bcl11b cKOs compared to controls and did not find Nrxn3 among the differentially expressed genes. To further validate this, we now quantified the Nrxn3 mRNA levels via qPCR in Bcl11b cKOs compared to controls and found no differences. These data are included in supplementary figure 5a of the revised manuscript.

-It appears that C1ql2 expression is much lower in the Nrxn123 KO mice. Since the authors are trying to test whether Nrxn3 is required for the correct targeting of C1ql2, this is a confounding factor. We can't really tell if what we are seeing is a "mistargeting" of C1ql2, loss of expression, or both. If the authors did a similar analysis to what they did in Figure 1 where they looked at the synaptic localization of C1ql2 (and quantified it) that could provide more evidence to support or refute the "mistargeting" claim.

Please also see response to reviewer #3 point 5 of public reviews. To exclude that reduction of fluorescence intensity of C1ql2 at the SL in Nrxn123 KO mice is due to loss of C1ql2 expression, we examined the mRNA levels of C1ql2 in control and Nrxn123 mutants and found no changes (data are included in supplementary figure 6b of the revised manuscript), suggesting that C1ql2 gene expression is normal. The reduced C1ql2 fluorescence intensity at the MFS was first observed when non-binding C1ql2 variant K262E was introduced to Bcl11b cKO mice that lack endogenous C1ql2 (Fig.6). In these experiments, we found that despite the overall high protein levels of C1ql2.K262E in the hippocampus (Fig. 6c), its fluorescence intensity at the SL was significantly reduced compared to WT C1ql2 (Fig. 6d-e). The remaining C1ql2.K262E signal in the SL was equally distributed and in a punctate form, similar to WT C1ql2. Together, this indicates that the loss of C1ql2-Nrxn3 interaction interferes with the localization of C1ql2 along the MFS, but not with expression of C1ql2. Of course, this does not exclude that additional mechanisms regulate C1ql2 localization at the synapse, as both the mutant C1ql2 in Bcl11b cKO and the endogenous C1ql2 in Nrxn123 cKO show residual immunofluorescence at the SL.

We note here that we have not previously quantified the co-localization of C1ql2 with individual synapses. C1ql2 is a secreted molecule that localizes at the MFS synaptic cleft. However, not much is known about the number of MFS that are positive for C1ql2 nor about the mechanisms regulating C1ql2 targeting, transport, and secretion to the MFS. Whether C1ql2 interaction with Nrxn3 is necessary for the protection of C1ql2 from degradation, its surface presentation and transport or stabilization to the synapse is currently unclear. Upon revision of our manuscript, we realized that we might have overstated this particular finding and have now rephrased the specific parts within the results to appropriately describe the observation and have also included a sentence in the discussion referring to the lack of understanding of the mechanism behind this observation.

-Title of Figure S5 is "Nrxn KO perturbs C1ql2 localization and SV recruitment at the MFS", but there is no data on C1ql2 localization.

This issue has been fixed in the revised manusript.

-S5 should be labeled more clearly than just Cre+/-

This issue has been fixed in the revised manuscript.

References

Castillo, P.E., Malenka, R.C., Nicoll, R.A., 1997. Kainate receptors mediate a slow postsynaptic current in hippocampal CA3 neurons. Nature 388, 182–186. https://doi.org/10.1038/40645

De Bruyckere, E., Simon, R., Nestel, S., Heimrich, B., Kätzel, D., Egorov, A.V., Liu, P., Jenkins, N.A., Copeland, N.G., Schwegler, H., Draguhn, A., Britsch, S., 2018. Stability and Function of Hippocampal Mossy Fiber Synapses Depend on Bcl11b/Ctip2. Front. Mol. Neurosci. 11.https://doi.org/10.3389/fnmol.2018.00103

Kaeser, P.S., Regehr, W.G., 2017. The readily releasable pool of synaptic vesicles. Curr. Opin. Neurobiol. 43, 63–70. https://doi.org/10.1016/j.conb.2016.12.012

Lerma, J., 2003. Roles and rules of kainate receptors in synaptic transmission. Nat. Rev. Neurosci. 4, 481–495. https://doi.org/10.1038/nrn1118

Orlando, M., Dvorzhak, A., Bruentgens, F., Maglione, M., Rost, B.R., Sigrist, S.J., Breustedt, J., Schmitz, D., 2021. Recruitment of release sites underlies chemical presynaptic potentiation at hippocampal mossy fiber boutons. PLoS Biol. 19, e3001149.https://doi.org/10.1371/journal.pbio.3001149

Vandael, D., Borges-Merjane, C., Zhang, X., Jonas, P., 2020. Short-Term Plasticity at Hippocampal Mossy Fiber Synapses Is Induced by Natural Activity Patterns and Associated with Vesicle Pool Engram Formation. Neuron 107, 509-521.e7. https://doi.org/10.1016/j.neuron.2020.05.013